# Switching Temporary Teachers for Semi-Supervised Semantic Segmentation

**Jaemin Na**[1], **Jung-Woo Ha**[2], **Hyung Jin Chang**[3], **Dongyoon Han**[2†], **Wonjun Hwang**[1,2†]

Ajou University, Korea[1], NAVER AI Lab[2], University of Birmingham, UK[3]

`osial46@ajou.ac.kr, jungwoo.ha@navercorp.com, h.j.chang@bham.ac.uk`
`dongyoon.han@navercorp.com, wjhwang@ajou.ac.kr`

## Abstract

The teacher-student framework, prevalent in semi-supervised semantic segmentation, mainly employs the exponential moving average (EMA) to update a single teacher's weights based on the student's. However, EMA updates raise a problem in that the weights of the teacher and student are getting coupled, causing a potential performance bottleneck. Furthermore, this problem may become more severe when training with more complicated labels such as segmentation masks but with few annotated data. This paper introduces Dual Teacher, a simple yet effective approach that employs dual temporary teachers aiming to alleviate the coupling problem for the student. The temporary teachers work in shifts and are progressively improved, so consistently prevent the teacher and student from becoming excessively close. Specifically, the temporary teachers periodically take turns generating pseudo-labels to train a student model and maintain the distinct characteristics of the student model for each epoch. Consequently, Dual Teacher achieves competitive performance on the PASCAL VOC, Cityscapes, and ADE20K benchmarks with remarkably shorter training times than state-of-the-art methods. Moreover, we demonstrate that our approach is model-agnostic and compatible with both CNN- and Transformer-based models. Code is available at https://github.com/naver-ai/dual-teacher.

## 1   Introduction

Semantic segmentation has progressed along with the advance of deep neural networks as a critical component of computer vision tasks for visual understanding [8, 9, 55]. However, the prevailing approaches are primarily based on supervised learning methods and still require laborious and time-consuming manual annotations at the pixel-level. To alleviate this intrinsic challenge, semi-supervised semantic segmentation has recently come into the limelight requiring only a small amount of labels.

A crucial challenge in successful semi-supervised learning is how to acquire reliable and consistent labels from unlabeled data, particularly in more demanding tasks like semi-supervised segmentation. Pseudo-labeling [22], the most popular approach, assigns the off-the-shelf model's class predictions as labels for unlabeled data for training. Another seminal work, a teacher-student framework [36], is widely adopted as it provides robust results in numerous semi-supervised learning tasks [41, 26, 13]. Despite the tremendous success, it is problematic that the weights of the teacher and student models are inevitably coupled as the teacher weights are updated using an exponential moving average (EMA) of the student model weights. This notable issue is called the *teacher-student coupling* problem [17] and causes the performance bottleneck for the student model because the EMA teacher exhibits similar performance and perspective to the student model in traditional teacher-student framework [36]. Furthermore, when the student model produces biased predictions over a few samples, training

---

[†]Corresponding authors.

37th Conference on Neural Information Processing Systems (NeurIPS 2023).

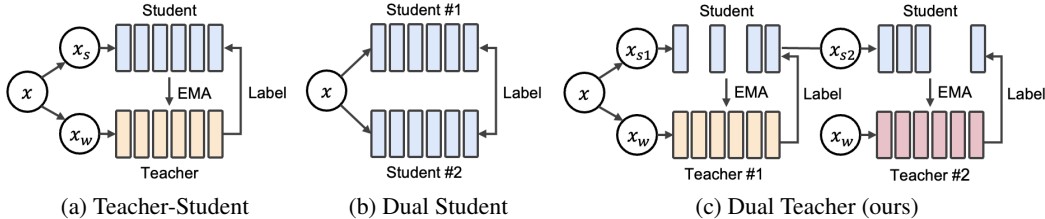

|  (a) Teacher-Student | (b) Dual Student | (c) Dual Teacher (ours) |

Figure 1: **Semi-supervised semantic segmentation frameworks.** (a) A teacher model generates pseudo-labels to train a student model and is updated as an exponential moving average (EMA) of the student model weights. (b) Dual Student involves two distinct student models that also serve as teachers to each other. (c) Our Dual Teacher employs dual temporary teachers that switch alternately during training.

with a single EMA teacher is prone to accumulate errors and provide incorrect pseudo-labels to the student model, which causes the irremediable misclassification (*i.e.,* confirmation bias [1]). One straightforward approach [17, 16, 6] to mitigate the issues is to employ explicit ensemble models with different initializations instead of counting on a single student model. However, incorporating another deep neural network is not preferred due to the heavy computational demands and the inherent challenge of achieving scalability.

In this paper, we propose a novel *Dual Teacher* framework that comprises a single-student model and dual temporary teacher models to alternatively handle the coupling problem between the teacher and student network. Our framework, which is depicted in Figure 1, discourages the teacher and student models from converging and becoming similar to each other as training progresses. We feed the student model with different strongly augmented inputs for each epoch in training, thereby encouraging the construction of a student model with distinct characteristics at every epoch. At the same time, as we change the augmentation applied to the student model, we also switch the temporary teacher models assigned to it. This ensures that the temporary teacher models keep retaining the characteristics of the student model for each epoch. Additionally, we improve the robustness of the student model by enforcing consistent predictions between fractional models of the student model and the full teacher model. These sub-models contribute to an implicit ensemble, which is differentiated from the explicit ensemble.

Ultimately, our approach achieves competitive performance with state-of-the-art methods [41, 15, 24] while requiring much shorter training times and fewer parameters. We demonstrate the superiority of Dual Teacher under the semi-supervised semantic segmentation protocol using public benchmarks, including PASCAL VOC [9] and Cityscapes [8]. We further prove the scalability of our method using the large-scale dataset, ADE20K [55], and have made our experimental protocols available to the public. We conduct extensive experiments not only with ResNet-50/-101 but also with Transformer-based SegFormer [43].

## 2 Related Work

**Semi-supervised Learning.** Semi-supervised learning (SSL) methods benefit significantly from consistency regularization [2, 31], which encourages a model to produce the same prediction when the input is perturbed, and pseudo-labeling [22] which uses a model's prediction as a label to train against. Recently, semi-supervised learning methods [2, 3, 32] have shown that consistency regularization and pseudo-labeling can work together harmoniously. Most of all, FixMatch [32] generates predicted labels from weakly-augmented unlabeled images and exploits them as pseudo-labels when strongly augmented versions of the same images are fed into the identical model. This simple yet effective approach has been widely used in recent studies when leveraging unlabeled data in SSL.

**Semi-supervised Semantic Segmentation.** Recent success in semi-supervised semantic segmentation approaches can be attributed to the effective incorporation of consistent regularization and pseudo-labeling. For example, CPS [6] and GCT [16] construct two parallel segmentation networks with the same structure and impose consistency on two networks with different initializations for the same input. Instead of using two different networks, which is burdensome to train, the latest works [41, 26, 15] adopt a teacher-student framework in which only the student model is trained, and the teacher model is updated through the EMA of the student model. On top of this simple

framework, GTA-Seg [15], U$^2$PL [41], and ReCo [24] introduce additional representation heads, and GTA-Seg further attaches a teacher assistant model the same size as the student model.

On the other hand, simple and effective data augmentations [49, 29, 50] that yield label-preserving perturbations have played a critical role in consistency regularization for semi-supervised semantic segmentation [6, 41, 24, 46]. Moreover, AEL [13] and AugSeg [53] contribute to performance improvement by introducing more advanced adaptive data augmentation techniques. However, unlike the previous studies, we achieve comparable performance by leveraging basic augmentations within our novel teacher-student framework without any lateral networks or sophisticated data augmentations. Meanwhile, PS-MT [26] presents an ensemble strategy that leverages multiple teachers to enhance the segmentation accuracy of unlabeled images. In contrast, rather than ensembling them, we employ the multiple teacher models in a sequential and independent manner. This approach ensures that each teacher model, with its diverse characteristics, distinctly contributes to the student model.

**Ensemble Learning.** Network ensembling [19] gains popularity for improving model predictions by combining the outputs of diverse models. Recent works [21, 52, 42] have successfully improved performance by model ensembling; that is, the individual models are trained to maximize their diversity so that the ensemble models are less prone to overfitting. Although these straightforward explicit ensemble approaches have shown significant performance improvements, they share the downside of bearing a burden of computation and memory cost. The impact of the explicit ensembling has also been indirectly exploited to train a single network through dropout [33], dropconnect [40], or stochastic depth [14] by activating only a subset of the network, and thus the complete network can be seen as an implicit ensemble of trained sub-networks [34, 37]. Of particular interest is that Temporal Ensembling [20] forms a consistent prediction of the unlabeled data using the outputs of the network being trained at different epochs under different regularization and input augmentations. Taking a step further with this approach, Mean Teacher [36] updates the teacher model with EMA weights of the student model instead of sharing weights with the student model as an ensembling perspective. Although this teacher-student framework has exhibited a significant impact, one critical problem is the coupling problem, wherein tightly coupled weights deteriorate performance. Alternatively, we address this problem by introducing dual temporary EMA teachers utilizing temporary and implicit ensemble perspectives.

## 3 Method

### 3.1 Preliminary

**Teacher-student framework.** The teacher-student framework [36] is a popular approach in semi-supervised learning that leverages unlabeled data to enhance model performance. As described in Figure 1 (a), it comprises a teacher and a student model, where each model includes an encoder and a decoder. In semi-supervised semantic segmentation, the student model is trained with a few labeled data as follows:

$$\mathcal{L}_{sup} = \frac{1}{\mathcal{B}_l} \sum_{i=1}^{\mathcal{B}_l} \frac{1}{\mathcal{H} \cdot \mathcal{W}} \sum_{j=1}^{\mathcal{H} \cdot \mathcal{W}} \mathcal{L}_{ce}(p_{ij}^l, y_{ij}^l), \tag{1}$$

where $\mathcal{B}_l$ denotes the number of labeled images $x_{ij}^l$ with a resolution of $\mathcal{H} \times \mathcal{W}$ in a training batch, and the $\mathcal{L}_{ce}$ is a pixel-wise cross-entropy loss applied to every pixel $j$ on $i$-th labeled image. The $p_{ij}^l$ indicates prediction for the labeled images of the student model, and $y_{ij}^l$ is ground-truth labels of corresponding labeled images. In this work, we include this supervised loss for the student model, and unsupervised loss for unlabeled data is addressed in Section 3.2.

The teacher model's parameters $\theta_t$ are updated with an EMA of the student model's parameters $\theta_s$ as:

$$\theta_t \leftarrow \alpha \theta_t + (1 - \alpha)\theta_s, \tag{2}$$

where $\alpha$ is a smoothing coefficient hyperparameter.

**Semi-supervised semantic segmentation.** In semi-supervised semantic segmentation, sophisticated data augmentations and network architectures are essential for handling pixel-level labels with increased degrees of freedom. However, due to the limited number of training instances compared to the abundance of mask labels, a more diversified student model should be involved than semi-supervised

classification. Furthermore, we argue that incorporating cumbersome data augmentations [13, 53] and additional heads [41, 15, 24] may impair efficiency, limiting the full potential of its benefits.

**Revisiting Temporal Ensembling.** Our work is inspired by Temporal Ensembling [20], which pursues to enjoy an indirect ensemble (*i.e.,* implicit ensemble) effect within a single model by ensembling predictions obtained from different regularizations for each epoch. This training approach accumulates the network's predictions into ensemble predictions after every training epoch. This process has been devalued for its slow training pace due to updating only once per epoch; however, we revalue it as a favorable method to inject diversity into our student model to address the coupling problem.

## 3.2 Dual Temporary Teacher

In the context of semi-supervised semantic segmentation, our goal is to mitigate the coupling problem [17] caused by the single EMA update routine employed in the conventional teacher-student framework. Among the critical factors [17, 32] for improving the teacher model, one we mainly take notice of is the diversity of the student models, as noted in prior works [17, 25, 44]. We strive to prevent the teacher and student models from becoming too similar to each other as training goes on by establishing a virtuous cycle in which the diversity of the student model enhances the teacher model. In turn, the improved teacher preserves students' diverse perspectives and effectively guides the student model. Since the student updates each EMA teacher, the teachers get enhanced as simultaneously diversifying the student due to the ensemble effect [19, 36]. The strengthened teacher model can subsequently provide more refined supervisory signals to the student model.

From this perspective, we introduce a simple yet effective framework - Dual Teacher: an approach that temporary EMA teacher models alternate in generating pseudo-labels to guide a student model; concurrently, the student model updates the weights of the teacher models with an exponential moving average of its own weights. We claim that introducing additional EMA teachers facilitates the diversification of students by providing distinct and varied supervision. As described in Figure 1 (c), Dual Teacher consists of a pair of teacher-student models, where the two temporary teacher models are switched every epoch during training to teach the single student model.

Similar to prior studies [32, 26, 41], we adopt strongly-augmented images as inputs for the student model and weakly-augmented images for the teacher models to ensure reliable pseudo-labels. However, instead of the traditional teacher-student framework that relies on a single permanent teacher model, we introduce dual temporary teachers alternately activated per epoch, providing diversified guidance to the student model. The temporary EMA teachers capture the evolving temporal knowledge of the student model, performing like a temporal ensemble at different time steps.

**Strong augmentation pool**. To diversify the student model more, we go beyond leaning solely on the EMA teacher model, albeit with alternating dual teachers. To further ensure diversity, we provide variety by changing the type of strong augmentation applied to the student model when shifting the temporary teacher models per epoch. We achieve this by building a predefined but non-deterministic pool of strong augmentations, including both fine class-level and coarse region-level augmentations (*i.e.,* ClassMix [29] and CutMix [49])[1]. Each training epoch randomly samples one augmentation from the pool, constraining that the consecutive epoch does not use the same augmentation.

**Updating student and teachers.** In a nutshell, the use of temporary teacher models helps to introduce diversity into the student model, while the diverse characteristics acquired by the student model contribute to the enhancement of the teacher models. Formally, the objective functions for the student model are defined as follows:

$$\theta_s \leftarrow \theta_s + \eta \frac{\partial(\mathcal{L}_{sup} + \lambda_u \mathcal{L}_{unsup})}{\partial \theta_s}, \quad \mathcal{L}_{unsup} = \frac{1}{\mathcal{B}_u} \sum_{i=1}^{\mathcal{B}_u} \frac{1}{\mathcal{H} \cdot \mathcal{W}} \sum_{j=1}^{\mathcal{H} \cdot \mathcal{W}} \mathcal{L}_{ce}(p_{ij}^u, \hat{y}_{ij}^u), \quad (3)$$

where $\mathcal{B}_u$ denotes the number of unlabeled images in a training batch, an image size of $\mathcal{H} \times \mathcal{W}$, $j^{th}$ pixel on the $i^{th}$ image, the student model's prediction $p_{ij}^u$ with unlabeled input applied strong augmentation, and the corresponding pseudo-label $\hat{y}_{ij}^u$ from the teacher models. The $\lambda_u$ is a scalar hyper-parameter to adjust the unsupervised loss weight.

---

[1]The coarse region-level augmentation helps the model exploit contextual information, while the fine class-level augmentation enables differentiation between objects based on their unique properties.

One of the temporary teachers is activated alternately at each epoch and retains the student model's characteristics through the EMA weights. The $k$-th temporary teacher $k \in \{1, ..., t_n\}$ is alternately switched at each epoch; the parameters of $k$-th temporary teacher $\theta_t^k$ are updated through EMA based on the student parameters $\theta_s$ by Eq. (2).

### 3.3 Implicit Consistency Learning

We introduce another implicit ensemble learning from the viewpoint of consistency regularization to enhance our student model. Motivated by [14, 38, 10], we encourage a subset of layers to be active in the student model and all these sub-models to make consistent predictions. In contrast to the previous studies [38, 2] that enforce consistent predictions between the full model and sub-models within the same model, we impose the consistency between the sub-models of the student model and the full teacher models. We construct the sub-models within the student model via versatile stochastic depth [14], applicable from CNN models to transformer-based models. Therefore, the student model is trained through the following objective functions:

$$\mathcal{L}_{cons} = \frac{1}{\mathcal{B}} \sum_{i=1}^{\mathcal{B}} \frac{1}{\mathcal{H} \cdot \mathcal{W}} \sum_{j=1}^{\mathcal{H} \cdot \mathcal{W}} \mathcal{L}_{ce}(p_{ij}(\tilde{\theta}_s), \hat{y}_{ij}(\theta_t)), \tag{4}$$

where $\tilde{\theta}_s$ denotes the parameters of sub-models of student model with a drop rate $\tau$, and $\mathcal{B}$ is the number of images in a training batch. Note that we only apply weak augmentations to inputs for both teacher and student models, unlike in Eq. 3, where we feed strong augmented inputs to the student model and weak augmented inputs to the teacher model, respectively. Finally, this loss cooperates with the $\mathcal{L}_{unsup}$ to update the student model.

## 4 Experiment

In this section, we demonstrate the efficacy of our proposed method by conducting comprehensive comparisons with state-of-the-art approaches on various publicly available benchmarks. Furthermore, we provide additional ablation studies to justify the significance and impact of our method.

### 4.1 Setup

**Datasets.** We evaluate our method on three public benchmarks in semantic segmentation, including the PASCAL VOC [9], Cityscapes [8], and ADE20K [55] datasets.

**PASCAL VOC** [9] is a widely-used semantic segmentation benchmark that consists of 20 object categories in the foreground and a background category. It includes 1,464 and 1,449 images in the training and validation set, respectively. To evaluate our method, we follow recent protocols [46, 45, 41] and validate it on both the original high-quality set and the blended set, which consist of 10,582 training images. For the blended set, we follow the common practice [47, 13, 6] and make use of the SBD [11] augmented set, comprising 9,118 training images with their corresponding annotations.

**Cityscapes** [8] is designed for semantic understanding of urban street scenes, including 30 classes, of which only 19 classes are used for scene parsing evaluation. There are 5,000 finely annotated images and 20,000 coarsely annotated images. The fine-annotated images are split into training, validation, and test sets with 2,975, 500, and 1,525 images, respectively. Following the prior works [13, 45, 41], we evaluate our method on $1/16$, $1/8$, $1/4$, and $1/2$ label partitions.

**ADE20K** [55] is a large-scale scene parsing dataset that covers 150 object and stuff categories. It consists of 25,574, 2,000, and 3,000 images in the training, validation, and test set, respectively. Due to the lack of standardized label partitions for semi-supervised segmentation scenarios, we randomly split the whole training set and construct $1/32$, $1/16$, $1/8$, $1/4$, and $1/2$ label partitions. We have publicly released these partitions on our GitHub repository.

**Model architectures.** We leverage various architectures to demonstrate the versatility of our model-agnostic approach. We adopt DeepLabv3+ [5] based on ResNet-50/-101 backbones [12] for PASCAL VOC, and ResNet-101 for Cityscapes. In addition, we employ DeepLabv2 [4] based on ResNet-101 for unsupervised domain adaptation task. Aside from the CNN backbones, we further validate our method with transformer-based architecture SegFormer [43] with the MiT-B1.

Table 1: Comparison of mIoU (%) with state-of-the-art methods on **PASCAL VOC 2012** under different partitions. Labeled images are sampled from the *original high-quality* training set. All methods are based on DeepLabv3+ with ResNet-50 backbone. We also report the total amount of trainable parameters in each method.

| Method | #Params | 1/16 (92) | 1/8 (183) | 1/4 (366) | 1/2 (732) | Full (1,464) |
|---|---|---|---|---|---|---|
| Supervised-only | 43.6M | 44.03 | 52.26 | 61.65 | 66.72 | 72.94 |
| PseudoSeg [56] [ICLR'21] | 43.6M | 54.89 | 61.88 | 64.85 | 70.42 | 71.00 |
| PC$^2$Seg [54] [ICCV'21] | 43.6M | 56.90 | 64.63 | 67.62 | 70.90 | 72.26 |
| AugSeg [53] [CVPR'23] | 43.6M | 64.22 | 72.17 | 76.17 | 77.40 | **78.82** |
| UniMatch [46] [CVPR'23] | 43.6M | **71.9** | 72.5 | 76.0 | 77.4 | 78.7 |
| Ours | 43.6M | 70.76 | **74.53** | **76.43** | **77.68** | 78.15 |

Table 2: Comparison of mIoU (%) with state-of-the-art methods on **PASCAL VOC 2012** under different partitions. Labeled images are sampled from the *blender* training set, which consists of 10,582 samples in total. All methods are based on DeepLabv3+ with ResNet-101 backbone.

| Method | #Params | 1/16 (662) | 1/8 (1,323) | 1/4 (2,646) | 1/2 (5,291) |
|---|---|---|---|---|---|
| Supervised-only | 62.6M | 67.87 | 71.55 | 75.80 | 77.13 |
| MT [36] [NeurIPS'17] | 62.6M | 70.51 | 71.53 | 73.02 | 76.58 |
| CPS [6] [CVPR'21] | 125.2M | 74.48 | 76.44 | 77.68 | 78.64 |
| AEL [13] [NeurIPS'21] | 62.6M | 77.20 | 77.57 | 78.06 | 80.29 |
| U$^2$PL [41] [CVPR'22] | 64.5M | 77.21 | 79.01 | 79.30 | 80.50 |
| GTA-Seg [15] [NeurIPS'22] | 127.1M | 77.82 | 80.47 | 80.57 | **81.01** |
| PCR [45] [NeurIPS'22] | 64.5M | 78.60 | 80.71 | 80.78 | 80.91 |
| Ours | 62.6M | **78.82** | **81.19** | **81.03** | 80.62 |

**Implementation details.** For the training on PASCAL VOC, we employ the stochastic gradient descent (SGD) optimizer with an initial learning rate of 0.001 and weight decay of 0.001. We set a batch size to 16 and run 80 epochs. Following the previous protocol [46], we use an image resolution of $321 \times 321$ for the original high-quality training set while configuring the blender set to $513 \times 513$. For Cityscapes, we also use SGD optimizer with an initial learning rate of 0.01, weight decay of 1e-4, batch size of 16, crop size of $769 \times 769$, and total training epochs of 200. For the training on the ADE20K, we follow the recipe of SegFormer [43, 7] and set the learning rate of 6e-5 and poly LR scheduler with a factor of 1.0. For a fair comparison, we validate the state-of-the-art methods [6, 41, 15, 24] using their official training codes with model architectures of SegFormer without any modifications. We train the methods with a batch size of 8 and crop size of $512 \times 512$ for the MiT-B1 backbone. Following the previous work [41], we progressively increase EMA weights up to 0.99 across all experiments.

**Evaluation.** In all experiments, we use the mean of Intersection over Union (mIoU) as a metric to evaluate the performance of semantic segmentation. For Cityscapes, we apply online hard example mining loss as the supervision loss and sliding window evaluation following the prior works [41, 53].

## 4.2 Comparison with State-of-the-Arts

**On PASCAL VOC.** In Table 1 and Table 2, we present the comparison results on validation set under different backbone networks (*i.e.,* ResNet-50/101) to validate the performance of our method. Table 1 demonstrates the effectiveness of our approach, consistently surpassing the performance of the baseline with the ResNet-50 backbone. Compared to the current state-of-the-art methods, AugSeg and UniMatch, our method yields superior results in three out of the five partitions. Except for the UniMatch [46], other studies have a higher training resolution than ours: 512 *vs.* 321. More details are provided in the supplementary material.

Table 2 shows the comparison results with the state-of-the-art methods based on ResNet-101. Compared to the baseline, our method achieves a performance improvement of +10.95%, +9.64%, +5.23%, and +3.49% for the label partitions $1/16$, $1/8$, $1/4$, and $1/2$, respectively. While a few numbers achieve marginal improvements over the state-of-the-art approaches, it is noteworthy that our simple approach is competitive with more complicated methods, such as GTA-Seg [15] and PCR [45], which rely on additional networks, and AEL [13], which employs more sophisticated augmentations.

Table 3: Comparison of mIoU (%) with state-of-the-art methods on **Cityscapes** under different partitions. All methods are based on DeepLabv3+ with ResNet-101 backbone.

| Method | #Params | 1/16 (186) | 1/8 (372) | 1/4 (744) | 1/2 (1,488) |
|---|---|---|---|---|---|
| Supervised-only | 62.6M | 65.74 | 72.53 | 74.43 | 77.83 |
| CPS [6] [CVPR'21] | 125.2M | 74.72 | 77.62 | 79.21 | 80.21 |
| AEL [13] [NeurIPS'21] | 62.6M | 75.83 | 77.9 | 79.01 | 80.28 |
| PS-MT [26] [CVPR'22] | 62.6M | – | 76.9 | 77.6 | 79.09 |
| U$^2$PL [41] [CVPR'22] | 64.5M | 74.9 | 76.48 | 78.51 | 79.12 |
| PCR [45] [NeurIPS'22] | 64.5M | 73.41 | 76.31 | 78.4 | 79.11 |
| UniMatch [46] [CVPR'23] | 62.6M | 76.6 | 77.9 | 79.2 | 79.5 |
| Ours | 62.6M | **76.81** | **78.4** | **79.46** | **80.52** |

Table 4: Comparison of mIoU (%) with state-of-the-art methods on **ADE20K** under different partitions. All results are reported with single-scale inference on SegFormer with MiT-B1 backbone. †: Results of our re-implementation according to each official code.

| Method | #Params | 1/32 (631) | 1/16 (1,263) | 1/8 (2,526) | 1/4 (5,052) | 1/2 (10,105) |
|---|---|---|---|---|---|---|
| Supervised-only | 13.7M | 15.15 | 22.32 | 25.90 | 29.53 | 32.57 |
| CPS† [6] [CVPR'21] | 27.5M | 18.79 | 22.27 | 27.89 | 32.41 | 36.89 |
| U$^2$PL† [41] [CVPR'22] | 15.0M | 18.54 | 23.64 | 28.11 | 33.52 | 36.01 |
| GTA-Seg† [15] [NeurIPS'22] | 28.8M | 19.52 | 23.09 | 26.86 | 27.85 | 29.99 |
| ReCo† [24] [ICLR'22] | 14.4M | 21.11 | 23.70 | 27.74 | 30.17 | 35.54 |
| Ours | 13.7M | **22.60** | **26.01** | **30.60** | **33.74** | **37.30** |

**On Cityscapes.** Table 3 reports comparison results of our method against several existing methods on the Cityscapes validation set. Our method yields improvements of +11.07%, +5.87%, +5.03%, and +2.69% over the baseline in the $1/16$, $1/8$, $1/4$, and $1/2$ label partitions, respectively, demonstrating its effectiveness, particularly in the scarce-label setting. Notably, our method outperforms CPS [6] on $1/16$ partition, achieving over 2% higher mIoU despite using only half as many trainable parameters. In addition, we achieve superior performance across all the label partitions compared to AEL [13], a method that introduces more advanced augmentations – adaptive Copy-Paste and adaptive CutMix.

**On ADE20K.** We further validate the effectiveness of our approach using SegFormer [43] in Table 4, which unifies Transformer encoder and a lightweight multi-layer perception (MLP) decoder on ADE20K. Since there are no standard label partitions of ADE20K for semi-supervised semantic segmentation, we construct partitions by sampling randomly from the entire training data. For a fair comparison, we re-implement the current state-of-the-art methods according to their official codes under the same network architecture (*i.e.,* SegFormer with MiT-B1 backbone). Our method consistently outperforms the supervised baseline by +4.86%, +3.69%, +4.70%, +4.21%, and +4.73% in the $1/32$, $1/16$, $1/8$, $1/4$, and $1/2$ partitions, respectively. It is noteworthy that we achieve competitive performance despite using fewer trainable parameters compared to other methods.

### 4.3 Empirical Studies

**Analysis of diversity.** To demonstrate the coupling problem, we report the prediction distance between the teacher and student models. Here, we compute the mean square error of predictions between two models to measure prediction distance. As depicted in Figure 2 (a), the predictions of the EMA teacher are very close to those of the student model in the conventional teacher-student framework (denoted as Single Teacher). On the other hand, we observe that Dual Teacher consistently strives to maintain the disparity between the predictions of the teacher and student models during training. We argue that this approach allows the

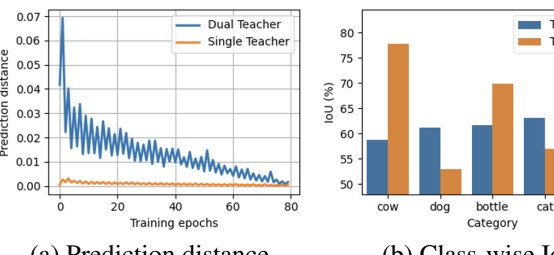

(a) Prediction distance     (b) Class-wise IoU

Figure 2: Results of Dual Teacher in training process on $1/16$ partitions of the PASCAL VOC *blender* set using ResNet-50.

Table 5: The performance relative to the number of teacher models and augmentations on $^1/_{16}$ splits of the PASCAL VOC *blender* set.

| Method | Single | Dual | Triple |
|--------|--------|------|--------|
| $Aug \times 1$ | 73.63 | 75.34 | 75.30 |
| $Aug \times 2$ | 74.21 | **77.00** | 76.56 |
| $Aug \times 3$ | 74.49 | 76.25 | 75.94 |

Table 6: Analyzing the diversity of supervisions using prediction distances between teacher-student models. A larger STD denotes a more diversified teacher's supervision.

| Epoch | 1 - 5 | 6 - 10 | 11 - 15 | 16 - 20 |
|-------|-------|--------|---------|---------|
| Single | 0.0018±0.0010 | 0.0016±0.0005 | 0.0012±0.0004 | 0.0012±0.0004 |
| Dual | 0.0378±0.0209 | 0.0249±0.0095 | 0.0181±0.0074 | 0.0199±0.0059 |
| Triple | 0.0425±0.0224 | 0.0169±0.0020 | 0.0194±0.0040 | 0.0222±0.0024 |

Table 7: Performance analysis of different teaching strategies on **PASCAL VOC 2012** across various partitions. Labeled images are sampled from the *blender* set and all methods are based on DeepLabv3+ with ResNet-50.

| Method | 1/16 (186) | 1/8 (372) | 1/4 (744) | 1/2 (1,488) |
|--------|------------|-----------|-----------|-------------|
| Switching teachers | 77.00 | 79.23 | 78.94 | 80.18 |
| Ensembling teachers | 73.25 | 75.99 | 76.41 | 77.34 |

student model to become more generalized, preventing it from learning towards a particular bias. This result supports our conjecture that the EMA teacher model is tightly coupled with the student model, which may incur performance degradation.

In Figure 2 (b), we investigate the distinct characteristics of the two temporary teacher models in Dual Teacher. Here, we select the top-5 categories with the most significant differences in class-wise IoU between dual temporary teacher models at the 10th epoch. Although the temporary teacher models have similar mIoU of 66.48% and 66.01% in the middle of the training process, we observe significant performance differences across each category. This observation indicates that our dual temporary teachers produce distinct supervisions and teach the student model complementarily.

**Impact of the number of teachers and augmentations.** In this study, we investigate our method by extending the number of temporary teachers from Dual Teacher to Triple Teacher while also expanding the number of augmentations. In Table 5, the first row shows the performance according to the number of teacher models by applying only ClassMix [29], which is specialized for semantic segmentation. We observe that employing two or more temporary teachers outperforms a single permanent teacher. This finding highlights the effectiveness of our approach that leverages teacher switching. In the second row, we begin to add CutMix [49] to our augmentation pool and utilize two different augmentations. We observe that the Dual Teacher achieves 2.79% improvement over the Single Teacher, and the Triple Teacher shows saturated performance compared to the Dual Teacher. It confirms that enriching strong augmentation types in addition to teacher switching is beneficial. In the last row, we incorporate three different augmentations by appending MixUp [50]. In this case, the results do not significantly outperform the counterparts of using two augmentations, regardless of the number of teachers. Ultimately, we draw a conclusion that leveraging both dual teachers and augmentations in our framework is the most effective.

**Dual teachers *vs.* Triple teachers.** In Table 6, we present a comparative analysis of the prediction distances between teacher and student models across several frameworks. Notably, the Dual Teacher exhibits more pronounced fluctuations in distances, as evidenced by its higher standard deviations (STDs). Although the trend shows a similar decline for both Dual Teacher and Triple Teacher across epochs, we observe a consistently lower STD for Triple Teacher. This suggests that Dual Teacher demonstrates greater variability in prediction distance across epochs, often varying more than double that of the Triple Teacher. We posit that since the three augmentation methods we used are probably not orthogonal, teacher models may tend to transfer similar characteristics to the student model due to the diminished diversity among teacher models. We presume the diversity inherent in teacher models is intrinsically connected to a student model's generalization capability, and our empirical backup supports it (see Table 5). Given the noticeable variations in prediction distances, we contend that teacher diversity is instrumental in amplifying the efficacy of Dual Teacher. As a result, teacher diversity is highly likely to contribute to enhanced performance for Dual Teacher, preventing the student model from learning towards a specific teacher bias.

**Switching teachers *vs.* Ensembling teachers.** One prevalent approach to leveraging dual teachers involves the ensembling of teacher models, which then simultaneously guide a student model [26]. In this teacher ensembling strategy, both teacher models consistently guide the student model throughout

Table 9: Comparison of mIoU (%) with the previous **unsupervised domain adaptation** methods on **Synthetic-to-Real: GTA5→Cityscapes**. All methods are evaluated on DeepLabv2 with ResNet-101.

| Method | Road | SW | Build. | Wall | Fence | Pole | TL | TS | Veg. | Terr. | Sky | PR | Rider | Car | Truck | Bus | Train | Motor | Bike | mIoU |
|---|---|---|---|---|---|---|---|---|---|---|---|---|---|---|---|---|---|---|---|---|
| Source-only | 63.31 | 15.65 | 59.39 | 8.56 | 15.17 | 18.31 | 26.94 | 15 | 80.46 | 15.25 | 72.97 | 51.04 | 17.67 | 59.68 | 28.19 | 33.07 | 3.53 | 23.21 | 16.73 | 32.85 |
| LTIR [18] | 92.9 | 55.0 | 85.3 | 34.2 | 31.1 | 34.9 | 40.7 | 34.0 | 85.2 | 40.1 | 87.1 | 61.0 | 31.1 | 82.5 | 32.3 | 42.9 | 0.3 | **36.4** | 46.1 | 50.2 |
| FDA [48] | 92.5 | 53.3 | 82.4 | 26.5 | 27.6 | **36.4** | 40.6 | 38.9 | 82.3 | 39.8 | 78.0 | 62.6 | 34.4 | 84.9 | 34.1 | 53.1 | 16.9 | 27.7 | 46.4 | 50.45 |
| PIT [27] | 87.5 | 43.4 | 78.8 | 31.2 | 30.2 | 36.3 | 39.9 | 42.0 | 79.2 | 37.1 | 79.3 | **65.3** | **37.5** | 83.2 | 36.0 | 45.6 | **25.7** | 23.5 | **49.9** | 50.6 |
| IAST [28] | 93.8 | 57.8 | 85.1 | **39.5** | 26.7 | 26.2 | **43.1** | 34.7 | 84.9 | 32.9 | **88.0** | 62.6 | 29.0 | 87.3 | 39.2 | 49.6 | 23.2 | 34.7 | 39.6 | 51.5 |
| Single Teacher | 93.34 | 56.81 | 86.28 | 34.63 | **32.27** | 32.99 | 39.21 | **43.93** | 86.72 | **45.11** | 85.81 | 63.05 | 21.28 | 86.76 | 38.66 | 50.05 | 0.0 | 13.8 | 33.7 | 49.71 |
| Dual Teacher | **93.93** | 56.88 | **86.47** | 35.49 | 31.57 | 35.12 | 37.78 | 43.43 | 86.02 | 43.54 | **88.06** | 62.14 | 28.49 | **88.59** | **50.14** | **54.2** | 0.18 | 35.75 | 49.82 | **53.03** |

all epochs. Therefore, this approach could limit the diversity of the student model, leading to weaker teacher models updated via EMA from the student model. On the other hand, our approach is differentiated by using a single switchable teacher model to instruct the student model in each epoch with more diversity. This ensures that our method produces a uniquely distinct student model for every epoch. To guarantee diversity among our dual temporary teachers, we update the teacher model from a student model trained with different augmentations for each epoch. Finally, Table 7 shows ensembling two teacher models reduces the diversity and significantly degrades performance in our method. We ensemble the outputs of the two teacher models in equal proportions, following the previous protocol [26] in this experiment.

**Impact of implicit consistency learning.** Table 8 shows the results of our study on implicit consistency learning with diverse decay strategies to determine the drop rate in our student model. Following the original work [14], we investigate linear and uniform decay rules with different drop rates. We use the fixed training regime to train ResNet-101 on Cityscapes

Table 8: The performance under different decay strategies and drop rate $\tau$ for implicit consistency.

| Decay Strategy | 0.1 | 0.2 | 0.3 | 0.4 |
|---|---|---|---|---|
| Linear decay | 75.87 | 76.55 | **76.63** | 75.74 |
| Uniform decay | **76.81** | 72.02 | 72.30 | 71.69 |

under $^1/_{16}$ label partitions. Compared to the baseline model (75.85 mIoU) trained without implicit consistency, the linear decay yields mostly stable performance improvements; but the performance degradation occurs at a drop rate of 0.4. On the other hand, uniform decay brings a notable enhancement at a drop rate of 0.1. Nevertheless, as the drop rate exceeds 0.1, performance gets diminished. Uniform decay drops early layers more than linear decay, which imposes more significant challenges. We claim this challenging aspect enhances an implicit ensemble learning effect. (also evidenced by the highest number, 76.81 mIoU); thereby, we adopt the uniform decay rule across all the experiments.

**Performance *vs*. Training time.** Here, we showcase the striking training efficiency of our method by comparing the trade-off between training time and mIoU against the current state-of-the-art methods on ADE20K. For a fair comparison, we run 200 epochs across all the methods. As described in Figure 3, our approach achieves competitive performance with less training time than other methods. Particularly, our method outperforms GTA-Seg, achieving 4.01% higher mIoU, requiring slightly less training time. In addition, we attain a notable improvement of 5.86% in mIoU with training time more than twice as fast as U$^2$PL. Surprisingly, our method attains comparable mIoU even with 12x faster than ReCo, which is based on pixel-level contrastive learning with sampling technique.

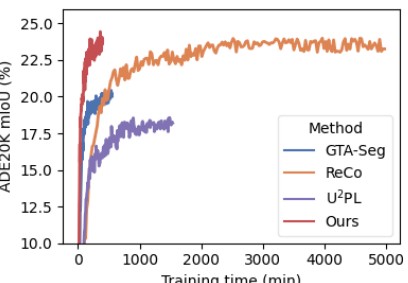

Figure 3: **mIoU (%) vs. training time**. Ours shows clear efficiency over the competing methods with a much improved final mIoU.

**Application to unsupervised domain adaptation.** We conduct extra experiments on unsupervised domain adaptation (UDA) task [39] for semantic segmentation. We apply our method on synthetic-to-real UDA scenarios using GTA5 [30] → Cityscapes benchmarks. In this experiment, we adopt DeepLabv2 [4] with ResNet-101, which is pretrained on MS COCO [23] following the previous UDA setting [39]. As reported in Table 9, our approach achieves comparable performance to the prior UDA methods and outperforms the conventional single-teacher approach. While our method does not demonstrate performance improvement in the "train" class, which suffers from a class-imbalance problem [35, 51], the overall performance supports the effectiveness of our method on the UDA task.

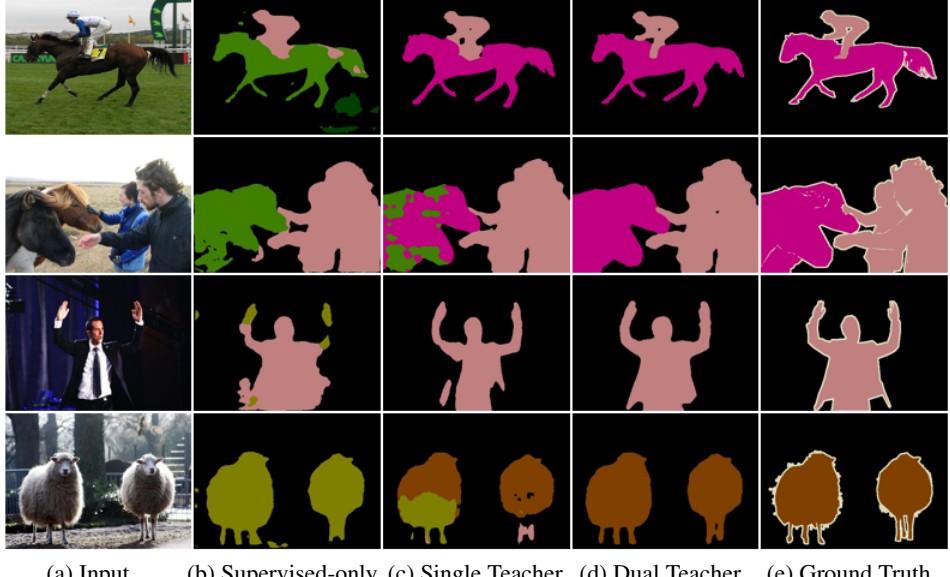

|          |                   |                  |                |                  |
|----------|-------------------|------------------|----------------|------------------|
| (a) Input | (b) Supervised-only | (c) Single Teacher | (d) Dual Teacher | (e) Ground Truth |

Figure 4: **Visualization of semantic segmentation results on PASCAL VOC.** We qualitatively compare the segmentation results of our Dual Teacher with other methods. All methods are evaluated with DeepLabV3+ on top of ResNet-101 under $^1/_{16}$ label partitions.

**Qualitative results.** We visualize the semantic segmentation results to compare our method with various methods in Figure 4. Column (b) shows the supervised-only method results, trained merely with a limited amount of labeled data, produce the least plausible segmentation results but make incorrect predictions for many pixels. For example, in the last row, "sheep" pixels are completely misclassified as "bird" in every pixel. Even though the Single Teacher yields slightly enhanced segmentation results, inaccurate pixels are still noticeable, as shown in column (c). Notably, in the first row, we can observe that Single Teacher produces inadequate results for pixels that contain different adjacent classes, such as person versus horse pixels. On the other hand, our Dual Teacher provides more sophisticated and accurate predictions compared to the Single Teacher.

## 5 Conclusion

In this paper, we have introduced a novel Dual Teacher framework to address the coupling problem in the widely used teacher-student framework caused by EMA updates. Instead of existing heavyweight solutions that rely on explicit ensemble, we have proposed a simple yet efficient approach from the perspective of implicit ensemble learning. Our approach not only achieved competitive performance compared with state-of-the-art methods but also required much less training time and parameters. We have demonstrated the superiority of our method over other competing approaches through extensive quantitative and qualitative comparisons. We hope that our method could serve as a valuable foundation for leveraging abundant unlabeled data in semantic segmentation tasks.

**Limitation.** Our focus was primarily on studying a more complicated segmentation task, but our method would apply to various other tasks, including classification or detection.

**Broader impact.** Advanced semantic segmentation methods might have a potential risk when applied to malicious image manipulation such as DeepFake. However, this work can contribute to more valuable applications in the real world.

**Acknowledgement.** This work was partially supported by the Institute of Information & communications Technology Planning & Evaluation (IITP) grant funded by the Korean government (MSIT) (RS-2023-00236245, Development of Perception/Planning AI SW for Seamless Autonomous Driving in Adverse Weather/Unstructured Environment) and (IITP-2023-No.RS-2023-00255968, Artificial Intelligence Convergence Innovation Human Resources Development).

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
