# Appendix

## A    Further Empirical Studies

**Switching teachers *vs.* Ensembling teachers (cont'd)** Our approach differs from teacher ensembling methods (*e.g.,* PS-MT [2]) in that only one switchable teacher guides the student per epoch, which allows our method to obtain a more distinct student model. We further compare the output predictions of two teachers of ours and PS-MT using class-wise IoU metric in Tables A1 and A2. We highlight the top-5 categories with "the most significant differences (ΔDiff)" in class-wise IoU between the teacher models. For a fair comparison, we compute class-wise IoU when both teachers have similar mIoU values (*e.g.,* ours: 66.48 / 66.01 *vs.* PS-MT: 67.09 / 67.37). The results demonstrate evidently distinct teacher supervisions, indicating that our method produces more diverse teacher models.

Table A1. Analysis of the top-5 categories with the most significant differences in our Dual Teacher.

| Category | cow | dog | cat | bottle | sheep |
|---|---|---|---|---|---|
| Teacher #1 | 58.73 | 61.24 | 61.66 | 63.16 | 79.55 |
| Teacher #2 | 77.70 | 52.98 | 69.81 | 56.91 | 75.97 |
| ΔDiff | 18.97 | 8.26 | 8.15 | 6.20 | 3.58 |

Table A2. Analysis of the top-5 categories with the most significant differences in PS-MT teachers.

| Category | bird | plant | table | chair | motorbike |
|---|---|---|---|---|---|
| Teacher #1 | 79.68 | 49.19 | 47.38 | 20.63 | 73.78 |
| Teacher #2 | 84.33 | 45.62 | 49.09 | 22.31 | 75.34 |
| ΔDiff | 4.65 | 3.57 | 1.71 | 1.68 | 1.67 |

We also summarize the prediction distance from 1 epoch to 20 epoch through the averages (AVG) and the corresponding standard deviations (STDs) (*i.e.,*, AVG $\pm$ STD):

Table A3. Comparison of the prediction distance between two types of teacher models across epochs.

| Epoch | 1 - 5 | 6 - 10 | 11 - 15 | 16 - 20 |
|---|---|---|---|---|
| Dual Teacher | 0.0378±0.0209 | 0.0249±0.0095 | 0.0181±0.0074 | 0.0199±0.0059 |
| PS-MT Teacher | 0.0067±0.0075 | 0.0012±0.0002 | 0.0010±0.0002 | 0.0009±0.0001 |

As reported in Table A3, PS-MT consistently shows lower distances than Dual Teacher shows. Specifically, the AVG/STD table shows that the mean of PS-MT is between 5 and 20 times smaller than Dual Teacher. The STD is similarly between 2 and over 50 times smaller. We presume such a significant difference between PS-MT and ours is because PS-MT always employs an ensemble of teacher networks (otherwise, ours switches teacher networks every epoch). The average between PS-MT's teachers (albeit they may have distinct characteristics) potentially becomes similar distances to the student at each epoch. As a result, PS-MT has a much small variance of the prediction distance at each epoch than Dual Teacher has. We argue this difference in prediction distances contributed to the final performance difference. Finally, we believe that the results above highlight the distinctions between our approach and PS-MT.

**More diverse augmentations.** We conduct an experiment to ascertain whether a more diverse augmentation can yield enhanced performance improvements. We exhibit it by adopting adaptive-CutMix proposed in AugSeg [7] instead of the basic CutMix [6]. We use PASCAL VOC 2012 with U$^2$PL splits with ResNet-50. As shown in Table A4, we observe mostly better results with a more sophisticated augmentation, adaptive CutMix. Adaptive CutMix does not beat CutMix for $1/4$ partitions; optimizing hyper-parameters would give improved results).

Table A4. Comparative analysis of performance based on different CutMix variations.

| Method | 1/16 (662) | 1/8 (1,323) | 1/4 (2,646) |
|---|---|---|---|
| Ours w/ Original-CutMix | 77.00 | 79.23 | 78.94 |
| Ours w/ Adaptive-CutMix | 78.66 | 79.59 | 78.52 |

## B    More Experimental Results

We further report additional quantitative results encompassing three different splits: *original high-quality set*, *blended set*, and *blended high-quality set*. Table A5 reports the comparison results on *original high-quality* training set under ResNet-50 backbone. Overall, we achieve competitive

results compared with state-of-the-art methods. UniMatch [4] employed a training resolution of 321, whereas other works opted for a higher resolution of 512. In scenarios characterized by a paucity of labels, specifically at partitions of $1/16$, $1/8$, and $1/4$, our method demonstrates superior efficacy at the training resolution of 321 in comparison to 512. Conversely, when the availability of labels is substantial, as observed at proportions of $1/2$ and Full, the resolution of 512 consistently yields better results. In experiments using a crop size of 321, we assign loss weights twice as high to labeled data as compared to unlabeled data, and use the same ratio of weights in all other experiments.

Table A5. Comparison of mIoU (%) with state-of-the-art methods on **PASCAL VOC 2012** under different partitions. Labeled images are sampled from the *original high-quality* training set. All methods are based on DeepLabv3+ with ResNet-50. We report performances for both crop sizes, 512 and 321.

| Method | #Params | 1/16 (92) | 1/8 (183) | 1/4 (366) | 1/2 (732) | Full (1,464) |
|---|---|---|---|---|---|---|
| Supervised-only | 43.6M | 44.03 | 52.26 | 61.65 | 66.72 | 72.94 |
| PseudoSeg [9] [ICLR'21] | 43.6M | 54.89 | 61.88 | 64.85 | 70.42 | 71.00 |
| PC$^2$Seg [8] [ICCV'21] | 43.6M | 56.90 | 64.63 | 67.62 | 70.90 | 72.26 |
| AugSeg [7] [CVPR'23] | 43.6M | 64.22 | 72.17 | 76.17 | 77.40 | **78.82** |
| UniMatch [4] [CVPR'23] | 43.6M | **71.9** | 72.5 | 76.0 | 77.4 | 78.7 |
| Ours \| **512×512** | 43.6M | 69.3 | 71.02 | 76.09 | **77.86** | 78.28 |
| Ours \| **321×321** | 43.6M | 70.76 | **74.53** | **76.43** | 77.68 | 78.15 |

Table A6 presents the comparison results with the state-of-the-art methods on *blended* set. Our method obtains superior performance in two of the three partitions examined. Remarkably, our method yields performance improvements of +2.3%, +2.48%, and +2.52% on the $1/16$, $1/8$, and $1/4$ partitions, respectively, compared to CPS-even though CPS has twice the number of parameters.

Table A6. Comparison of mIoU (%) with state-of-the-art methods on **PASCAL VOC 2012** under different partitions. Labeled images are sampled from the *blended* training set. All methods are based on ResNet-50.

| Method | #Params | 1/16 (662) | 1/8 (1,323) | 1/4 (2,646) |
|---|---|---|---|---|
| Supervised-only | 43.6M | 62.4 | 68.2 | 72.3 |
| CPS [1] [CVPR'21] | 87.3M | 71.98 | 73.67 | 74.90 |
| ST++[5] [CVPR'22] | 43.6M | 72.6 | 74.4 | 75.4 |
| PS-MT [2] [CVPR'22] | 43.6M | 72.83 | 75.7 | 76.43 |
| UniMatch [4] [CVPR'23] | 43.6M | 74.5 | 75.8 | 76.1 |
| AugSeg [7] [CVPR'23] | 43.6M | **74.66** | 75.99 | 77.16 |
| Ours | 43.6M | 74.28 | **76.15** | **77.42** |

In Table A7, we present the comparison results on *blended high-quality* set under different backbone networks (*i.e.,* ResNet-50/101) to evaluate the performance of our method. Upon comprehensive evaluation, it is discerned that our approach yields performance comparable to recent methods, irrespective of the underlying backbone architecture. We observe that the adoption of an advanced CutMix [7]) consistently outperforms the original CutMix.

Table A7. Comparison of mIoU (%) with state-of-the-art methods on **PASCAL VOC 2012** under different partitions. Labeled images are sampled from the *blended high-quality* training set [3]. All methods are based on DeepLabv3+ with ResNet-50/-101. †: Results using the adaptive-CutMix [7] instead of the original CutMix.

| Method | ResNet-50 | | | ResNet-101 | | |
|---|---|---|---|---|---|---|
| | 1/16 (186) | 1/8 (372) | 1/4 (744) | 1/16 (186) | 1/8 (372) | 1/4 (744) |
| Supervised-only | 67.7 | 71.9 | 74.5 | 70.6 | 75.0 | 76.5 |
| U$^2$PL [3] [CVPR'22] | 74.70 | 77.40 | 77.50 | 77.21 | 79.01 | 79.30 |
| AugSeg [7] [CVPR'23] | 77.28 | 78.27 | 78.24 | 79.29 | 81.46 | 80.5 |
| UniMatch [4] [CVPR'23] | 78.1 | 79.0 | **79.1** | **80.9** | **81.9** | 80.4 |
| Ours \| **Adaptive-CutMix** | 77.00 | 79.23 | 78.94 | 78.82 | 81.19 | **81.03** |
| Ours$^†$ \| **Original-CutMix** | **78.66** | **79.59** | 78.52 | 80.07 | 81.47 | 80.53 |