# OpenReview forum: "Switching Temporary Teachers for Semi-Supervised Semantic Segmentation"
_NeurIPS.cc/2023/Conference — NeurIPS 2023 poster_

### Official Review · Reviewer_WnDT · 2023-06-25

**Soundness:** 3 good
**Presentation:** 3 good
**Contribution:** 3 good
**Rating:** 6
**Confidence:** 5

**Summary:**

The authors argue that the teacher-student coupling problem leads to the performance bottleneck in semi-supervised semantic segmentation. To this end, this paper proposes a novel Dual Teacher framework. Specifically, it introduces two teachers that are switched every epoch during training to supervise the student model. By doing so, it manages an implicit ensemble and the improved model diversity do contribute to better segmentation performances. Experiments under various evaluation protocols demonstrate the efficacy of the proposed method.

**Strengths:**

1. This paper is well-written and easy to follow.
2. The teacher-student coupling problem is novel and worth studying in semi-supervised semantic segmentation.
3. Experiments are sufficient. Notably, the authors conduct experiments on ADE20k as well as using Transformer-based architectures to demonstrate the efficacy of the proposed method, which is not a common practice in the field of semi-supervised semantic segmentation.
4. The code is provided in the supplementary material.


**Weaknesses:**

1. More explanations are needed to Table 5. Specifically, why does the Triple Teacher framework perform worse than the Dual Teacher alternative? As mentioned, to alleviate the teacher-student coupling problem, diversity among models is crucial. *It may be helpful to plot the prediction distance of the Triple Teacher framework in Figure 2(a).* I guess 3 teachers should be more diverse than 2. If it is supported by empirical evidence, it is better to explain the underlying reason for the performance degradation in Table 5.
2. Missing ablations. It is relatively hard to evaluate the effectiveness of each component. It is better to start from the mean-teacher framework and add each component step by step.
3. Figure 1(c) seems to be incorrect. From the method section, $x_w$ should be fed into the teacher while $x_{s_1}$ and $x_{s_2}$ should be fed into the student.

**Questions:**

I do not have any questions. Please refer to the weaknesses section.


**Post Rebuttal Comments**

The reviewer appreciates the extra experiments a lot. Based on this evidence, it seems that the more diverse teachers we have, the better performance the student achieves, which is quite interesting. Also, as the teachers of PS-MT are much less diverse, the reviewer thinks that the student-teacher coupling problem has not been well addressed by PS-MT. Therefore, I am going to keep my initial rating: 6. The authors are supposed to revise their paper according to our dialog, especially the analysis of prediction distances.

**Limitations:**

The authors have discussed the limitations of this work in the submission.

---

> ### Author Rebuttal · Authors · 2023-08-09
>
> We appreciate the insightful comments and positive reviews. We address all the concerns below and will revise them in the final version.
>
> >Q1. Why does the Triple Teacher framework perform worse than the Dual Teacher alternative in Table 5? As mentioned, to alleviate the teacher-student coupling problem, diversity among models is crucial.
>
> A1. Thank you for the constructive comments. In our experiments, we adopted different types of augmentation, such as class-level and coarse region-level augmentations (i.e., ClassMix, and CutMix) to train our Dual Teacher. In addition, we introduced Mixup as a third augmentation strategy to train Triple Teacher because Mixup is another well-known augmentation method recently. Within this setting, we speculate that Triple Teacher could not outperform Dual Teacher because Mixup has a similar effect to the augmentations already used for the Dual Teacher. However, we expect the Triple Teacher could surpass the Dual Teacher if we can provide diverse views to the model, which is sufficiently different from the two augmentations preemptively applied to the dual teacher. We will study better diversifying augmentations for Triple Teacher.
>
> >Q2. It is better to start evaluating each component from the mean-teacher framework step by step.
>
> A2. In Table 5, 'Single Teacher' refers to the conventional Mean Teacher framework. Hence, in the first column, we can observe the performance effect when appending different augmentations to the basic Mean Teacher framework. Meanwhile, the first row shows the impact of the number of temporary teachers from the Mean Teacher framework. We will provide more detailed and clear explanations in the captions of the table.
>
> >Q3. Figure 1(c) seems to be incorrect. From the method section, $x_w$ should be fed into the teacher while $x_{s1}$ and $x_{s2}$ should be fed into the student.
>
> A3. Thank you for pointing this out. We will modify it to accurately represent that the weak augmented inputs ($x_w$) are fed to the teacher models and the strong augmented inputs ($x_{s1}$ and $x_{s2}$) are supplied to the student model, respectively.

---

> ### Comment · Reviewer_WnDT · 2023-08-15
> **Post Rebuttal Comments from Reviewer WnDT**
>
> I appreciate the rebuttal. However, I am not totally convinced, especially for the first concern.
> I would like to see the prediction distance of the Triple Teacher framework in Figure 2(a). In other words, I would like to check out whether there is coherence between the performance and the diversity of teachers.
>
> Moreover, I notice that Reviewer vns8 has pointed out that this paper is very similar to PS-MT, and I am not very convinced by the explanation. It is better to compare the diversity of the two methods.
>
> Therefore, I am considering lowering my rating.

---

> > ### Author Response · Authors · 2023-08-16
> >
> > We appreciate your insightful feedback and your comments. We would like to clarify your concern in the comments.
> >
> > **1. [1] Prediction distance of triple teacher framework and [2] prediction distance between ours and PS-MT.**
> >
> > Currently, we are conducting experiments aimed at making the prediction distances. Please understand that we need additional time to complete them. We gently remind you that there is no way to upload a figure at OpenReview so we will provide them in a table form.
> >
> > **2. Our main purposes of our method are [1] establishing a *decoupling* between teacher and student, [2] promoting *diversity* between teachers, [3] simple yet *efficient perturbation*, unlike PS-MT**.
> >
> > > [1] Main difference between PS-MT and ours lies in the method employed for teaching a pseudo label for a student.
> >
> > &emsp;&emsp;PS-MT&emsp;&emsp;|&emsp;Ours\
> > ------------------------------------\
> > $T_1$┐Ensemble&ensp;|&emsp;$T_1$&emsp;$T_2$\
> > $T_2$┘&emsp;&emsp;↓&emsp;&emsp;&ensp;|&emsp;↓&emsp;&emsp;↓\
> > &emsp;&emsp;&emsp;&ensp;$S$&emsp;&emsp;&ensp;|&emsp;$S$&nbsp;&nbsp;↔&nbsp;$S$\
> > ------------------------------------\
> >
> > As shown in the above figure, for establishing the *decoupling*, we switched teachers every epoch to guide the pseudo label to the student. However, PS-MT guided the pseudo label from the ensemble of teachers to the student.
> > Specifically, our method:\
> > [Epoch t+1 ]   $T_1$ teaches $S$ / then, $T_2$ is decoupled from $S$.\
> > [Epoch t+2 ]   $T_2$ teaches $S$ / then, $T_1$ is decoupled from $S$.
> >
> > PS-MT:\
> > [Epoch t+1 ] [$T_1$+$T_2$] teaches $S$ / then, Student cannot be decoupled from any Teachers.\
> > [Epoch t+2 ] [$T_1$+$T_2$] teaches $S$ / then, Student cannot be decoupled from any Teachers.
> >
> > From this sequential switching mechanism, we can achieve the *decoupling* of the student and teachers. It results in better performance compared with PS-MT.
> >
> > > [2] Main difference comes from diverse augmentations for students (e.g., $SA_1$ and $SA_2$).
> >
> > &emsp;&emsp;&emsp;PS-MT&emsp;&emsp;&emsp;&emsp;|&emsp;&emsp;&emsp;Ours\
> > ---------------------------------------------------------------------\
> > (WA→$T_1$)┐Ensemble&ensp;|&emsp;(WA→$T_1$)&emsp;(WA→$T_2$)\
> > (WA→$T_2$)┘&emsp;&emsp;↓&emsp;&emsp;&ensp;|&emsp;&emsp;&emsp;&emsp;↓&emsp;&emsp;&emsp;&emsp;&emsp;↓\
> > &emsp;&emsp;&emsp;&emsp;(SA→$S$)&emsp;&emsp;&ensp;|&ensp;($SA_1$→$S$)→&nbsp;($SA_2$→$S$)\
> > ----------------------------------------------------------------------\
> >
> > In the *diversity* of teachers, our method achieved better diversity of the teachers because of our switching teacher mechanism (supported by our different augmentations), unlike the simple ensemble of teachers. The proof of this concept has been done by the following experiments where we highlighted the top-5 categories with "the most significant differences ($\Delta$Diff)" in class-wise IoU between the teacher models.
> >
> > - Our method
> > |Category | cow | dog |  cat | bottle | sheep |
> > |:-:|:-:|:-:|:-:|:-:|:-:|
> > |**$\Delta$Diff of $T_1$ and $T_2$**|18.97|8.26|8.15|6.20|3.58|
> >
> > - PS-MT
> > | Category | bird | plant | table | chair | motorbike |
> > |:-:|:-:|:-:|:-:|:-:|:-:|
> > |**$\Delta$Diff of $T_1$ and $T_2$**|4.65|3.57|1.71|1.68|1.67|
> >
> > From this different augmentation mechanism, we can achieve the *diversity* of the dual teachers, contributing to better performance than PS-MT.
> >
> > > [3] Main difference on pertubation mechanism.
> >
> > &emsp;&emsp;&emsp;PS-MT&emsp;&emsp;&emsp;&emsp;|&emsp;&emsp;&emsp;Ours\
> > -------------------------------------------------------------------\
> > Feature pertubation&ensp;|&ensp; Layer pertubation\
> > &emsp;(T-VAT-based)&emsp;&emsp;&ensp;&nbsp;|&ensp;(Stochastic depth-based)\
> > --------------------------------------------------------------------\
> >
> > We employed stochastic depth-based layer perturbation to the student model for further consistency learning effect, and PS-MT used T-VAT-based feature perturbation. Specifically, our perturbation is simply made in the student model, while PS-MT's perturbation is made from the ensemble of teachers to the student. This difference allows us to achieve a simple yet *efficient perturbation* for faster training than PS-MT.

---

> > > ### Author Response · Authors · 2023-08-16
> > >
> > > We now provide **the prediction distance tables** to address the reviewer's concerns about comparisons: Dual Teacher vs. Triple Teacher and ours vs. PS-MT.
> > >
> > > **1. [1] Prediction distance of the triple teacher framework and [2] prediction distance between ours and PS-MT.**
> > >
> > > We gently remind you again that we cannot upload any figures at OpenReview, so we provide them using table formats. Please find the results below:
> > >
> > > **[1] Dual Teacher vs. Triple Teacher**
> > >
> > > We first provide the raw data on prediction distance from 1 epoch to 20 epoch (due to lack of space and for better readability):
> > >
> > > |Epoch|1|2|3|4|5|6|7|8|9|10|11|12|13|14|15|16|17|18|19|20|
> > > |---|---|---|---|---|---|---|---|---|---|---|---|---|---|---|---|---|---|---|---|---|
> > > |Dual Teacher|0.0417|0.0693|0.0221|0.0402|0.0155|0.0323|0.0163|0.0337|0.0131|0.0290|0.0135|0.0278|0.0135|0.0243|0.0115|0.0267|0.0150|0.0238|0.0126|0.0214|
> > > |Triple Teacher|0.0786|0.0488|0.0319|0.0315|0.0219|0.0191|0.0190|0.0149|0.0159|0.0156|0.0162|0.0168|0.0166|0.0223|0.0249|0.0244|0.0190|0.0216|0.0211|0.0247|
> > >
> > > We also summarize through the averages (AVG) and the corresponding standard deviations (STDs) in the above table for better readability (i.e., AVG ± STD):
> > >
> > > |Epoch|1-5|6-10|11-15|16-20|
> > > |---|---|---|---|---|
> > > |Dual Teacher|0.0378±0.0209|0.0249±0.0095|0.0181±0.0074|0.0199±0.0059|
> > > |Triple Teacher|0.0425±0.0224|0.0169±0.0020|0.0194±0.0040|0.0222±0.0024|
> > >
> > > As shown in the above tables, we can observe the distinct trends from the tables between dual and Triple Teacher:
> > >
> > > * **Dual Teacher has highly fluctuating distances (high STDs) compared with Triple Teacher.** More specifically,
> > >    - The average prediction distance decreases similarly for both Dual and Triple Teacher for every five epochs. However, we observe that the STD of Triple Teacher's prediction distances is usually less than half of Dual Teacher’s, which means Triple Teacher has less distance discrepancy at each epoch.
> > >    - As described in our response letter, since the three augmentation methods used in this paper are presumably not orthogonal to each other, teachers guide similar characteristics to students, which incurs low diversity of teachers.
> > >    - We claimed in the paper that teachers’ diversity might relate to the student’s generalization capability and achieved deviated performance: 77.00 (Dual Teacher) compared with 75.94 (Triple Teacher). We argue that the STD difference (i.e., teacher diversity) eventually contributes to Dual Teacher’s improved performance, preventing the student from learning towards a particular bias.
> > >    - Therefore, we can conclude that there is coherence between the performance and the diversity of teachers.
> > >
> > > **[2] Dual Teacher (ours) vs. PS-MT's teacher**
> > >
> > > We provide the raw data on each prediction distance of our method (Dual Teacher) and PS-MT at each epoch:
> > >
> > > |Epoch|1|2|3|4|5|6|7|8|9|10|11|12|13|14|15|16|17|18|19|20|
> > > |---|---|---|---|---|---|---|---|---|---|---|---|---|---|---|---|---|---|---|---|---|
> > > |Dual Teacher|0.0417|0.0693|0.0221|0.0402|0.0155|0.0323|0.0163|0.0337|0.0131|0.0290|0.0135|0.0278|0.0135|0.0243|0.0115|0.0267|0.0150|0.0238|0.0126|0.0214|
> > > |PS-MT's teacher|0.0193|0.0080|0.0029|0.0021|0.0012|0.0014|0.0010|0.0013|0.0010|0.0013|0.0009|0.0012|0.0009|0.0011|0.0008|0.0011|0.0008|0.0010|0.0008|0.0010|
> > >
> > > We also summarize the above table again for better readability through the averages (AVG) and the corresponding standard deviations (STDs) (i.e., AVG ± STD):
> > >
> > > |Epoch|1-5|6-10|11-15|16-20|
> > > |---|---|---|---|---|
> > > |Dual Teacher|0.0378±0.0209|0.0249±0.0095|0.0181±0.0074|0.0199±0.0059|
> > > |PS-MT’s teacher|0.0067±0.0075|0.0012±0.0002|0.0010±0.0002|0.0009±0.0001|
> > >
> > > * **PS-MT consistently shows lower distances than Dual Teacher shows.** More specifically,
> > >   - The AVG/STD table shows that the mean of PS-MT is between 5 and 20 times smaller than Dual Teacher. The STD is similarly between 2 and over 50 times smaller.
> > >   - We presume such a significant difference between PS-MT and ours is because PS-MT always employs an ensemble of teacher networks (otherwise, ours switches teacher networks every epoch). The average between PS-MT's teachers (albeit they may have distinct characteristics) potentially becomes similar distances to the student at each epoch.
> > >   - As a result, PS-MT has a much small variance of the prediction distance at each epoch than Dual Teacher has. We argue this difference in prediction distances contributed to the final performance difference.
> > >
> > > Finally, we believe that the results above highlight the distinctions between our approach and PS-MT.

---

> > > > ### Comment · Reviewer_WnDT · 2023-08-17
> > > >
> > > > The reviewer appreciates the extra experiments a lot. Based on this evidence, it seems that the more diverse teachers we have, the better performance the student achieves, which is quite interesting. Also, as the teachers of PS-MT are much less diverse, the reviewer thinks that the student-teacher coupling problem has not been well addressed by PS-MT. Therefore, I am going to keep my initial rating: 6. The authors are supposed to revise their paper according to our dialog, especially the analysis of prediction distances.

---

### Official Review · Reviewer_gkhQ · 2023-07-02

**Soundness:** 2 fair
**Presentation:** 2 fair
**Contribution:** 1 poor
**Rating:** 3
**Confidence:** 5

**Summary:**

This paper addresses the task of semi-supervised semantic segmentation. It proposes dual teachers that are alternatively updated to supervise a single student. Improvements are obtained compared with previous methods.

**Strengths:**

- Improvements over previous best methods on Pascal VOC and Cityscapes.
- The method is also validated in unsupervised domain adaptation.

**Weaknesses:**

- The practice of dual teachers has already been explored in PS-MT [1]. The teachers in PS-MT are also alternatively updated and then supervise the student by ensembling their predictions.
- Some latest methods should be compared in Table 2 and Table 3, such as AugSeg [2] and UniMatch [3]. The results are inferior to these methods in some splits. Besides, the authors should report the high-quality splits (92, 183, 366... labels) on Pascal. The Table 1 and Table 2 adopt the U2PL splits (choose from the high-quality images first), which are unfair for some methods, such as CPS and ST++ (randomly choose from the blended set).
- More ablation studies are required. For example, does the improvement come from more diverse spatial augmentations, i.e., CutMix and ClassMix?

[1] Perturbed and Strict Mean Teachers for Semi-supervised Semantic Segmentation, CVPR 2022.

[2] Augmentation Matters: A Simple-yet-Effective Approach to Semi-supervised Semantic Segmentation, CVPR 2023.

[3] Revisiting Weak-to-Strong Consistency in Semi-Supervised Semantic Segmentation, CVPR 2023.

**Questions:**

I see the authors upload the source code. Since several other works are reproduced for ADE20K, it will be more convincing and greatly appreciated if the authors can provide comprehensive experimental logs as well.

**Limitations:**

Yes, they have been discussed.

---

> ### Comment · Area_Chair_2zie · 2023-08-05
>
> This is an AC of #12117. First of all, thank you for your great contribution to NeurIPS 2023.
>
> I have discussed with PCs that if the authors can share experimental logs in the rebuttal period. We have concluded that such logs can be regarded as code in some way so the authors share them in a completely anonymized way.
>
> However, the author guideline (https://neurips.cc/Conferences/2023/PaperInformation/NeurIPS-FAQ) specifies that codes (i.e., experimental logs in our case) must be shared only with ACs, not directly with reviewers. Hence, the authors first have to send the logs to "me" so that I check them and provide a summary to you. As you may expect, this process will be quite difficult and take so long time without knowing which kind of experimental results you exactly want to see.
>
> For this reason, I ask you to be more specific on what you want to see from the logs. Then I will let the authors know them and provide the logs in a concise way.

---

> > ### Comment · Area_Chair_2zie · 2023-08-13
> >
> > The authors have provided training logs from runs of their method and previous work on ADE20K. Please let me know what you want to see from the logs so that I provide a brief summary on it. Also please note that we should take this information into consideration during the post-rebuttal discussion and for making the final decision.

---

> ### Author Rebuttal · Authors · 2023-08-09
>
> We appreciate the constructive reviews with suggestions. We address all the concerns and add new experiments that you suggested.
>
> >Q1. The practice of dual teachers has already been explored in PS-MT. The teachers in PS-MT are also alternatively updated and then supervise the student by ensembling their predictions.
>
> A1. We would like to highlight the following differences.
>
> 1. Our approach differs from PS-MT in that only one switchable teacher model teaches a student model per epoch, and this difference allows our method to obtain a more distinct student model for each epoch.
> 2. To ensure the diversity of our dual temporary teachers, we update the teacher model from a student model trained with different augmentations for each epoch.
>
> In PS-MT, two teachers contribute to training the student model in all epochs. They use the ensembled outputs of the two teachers in every epoch to provide better pseudo-labels to the student. This training approach is limited to obtaining a student that varies over epochs and results in teachers that are updated via EMA from the student not diversifying enough.
>
> However, in our method, only one of the temporary teachers involves teaching the student in each epoch. This approach was intended to diversify the teachers: our dual teachers are trained to have different perspectives using different types of augmentation, and only one of the teachers teaches the student with a distinct perspective at each epoch. We argue that ensembling teachers ensures consistent supervision for our method but diminishes diversified guidance for the student, which potentially hinders the student from becoming more generalized in the end.
>
> To provide the backup for our claim, we perform the following experiments to demonstrate that ensembling the outputs of the two teachers negatively affects obtaining diverse teachers in our method. Here, we ensemble the outputs of the two teachers in equal proportions, according to the ensemble method of PS-MT. Please check the results in the table below.
>
> |Method | 1/16 (186) | 1/8 (372) |  1/4 (744) | 1/2 (1,488) |
> |:-|:-:|:-:|:-:|:-:|
> |**Ours + Switching temporary teaching**|77.0|79.23|78.94|80.18|
> |**Ours + Ensemble teaching**|73.25|75.99|76.41|77.34|
>
> >Q2. Some latest CVPR2023 methods should be compared in Table 2 and Table 3, such as AugSeg and UniMatch.
>
> A2. We show performance comparisons of our method in Tables 2 and 3 as well as Table 1, including AugSeg [CVPR2023] and UniMatch [CVPR2023], and our method shows comparable results with the competing AugSeg and UniMatch on PASCAL VOC. During the rebuttal period, we get better accuracy using the adaptive CutMix employed in AugSeg. We will append these results to our tables. +: means we use adaptive CutMix instead of the original CutMix.
>
> - Comparison results on PASCAL VOC.
> |Method (ResNet-50)|1/16|1/8|1/4|Method (ResNet-101)|1/16|1/8|1/4|
> |:-:|:-:|:-:|:-:|:-:|:-:|:-:|:-:|
> |U$^2$PL|74.70|77.40|77.50|U$^2$PL|77.21|79.01|79.30|
> |AugSeg|77.28|78.27|78.24|AugSeg|79.29|81.46|80.5|
> |UniMatch|78.1|79.0|**79.1**|UniMatch|**80.9**|**81.9**|80.4|
> |**Ours**|77.0|79.23|78.94|**Ours**|78.82|81.19|**81.03**|
> |**Ours+**|**78.66**|**79.59**|78.52|**Ours+**|80.07|81.47|80.53|
>
> - Comparison results on Cityscapes.
> |Method | 1/16 (186) | 1/8 (372) |  1/4 (744) | 1/2 (1,488) |
> |:-:|:-:|:-:|:-:|:-:|
> |CPS|74.72|77.62|79.21|80.21|
> |AEL|75.83|77.9|79.01|80.28|
> |U$^2$PL|74.9|76.48|78.51|79.12|
> |PCR|73.41|76.31|78.4|79.11|
> |AugSeg|75.22|77.82|**79.56**|80.43|
> |UniMatch|76.6|77.9|79.2|79.5|
> |**Ours**|**76.81**|**78.4**|79.46|**80.52**|
>
> >Q3. Authors should report other splits (92, 183, 366... labels) on PASCAL VOC.
>
> A3. Thank you for the suggestion. In the submitted paper, we report performance following the experimental setup of U$^2$PL [CVPR-2022] but also provide the experimental results with the suggested splits on ResNet-50 here. We use the same image resolution in these experiments according to the standard training protocols (e.g., AugSeg).
> |Method | 1/16 (92) | 1/8 (183) |  1/4 (366) | 1/2 (732) | Full (1,464) |
> |:-:|:-:|:-:|:-:|:-:|:-:|
> |Supervised|44.03|52.26|61.65|66.72|73|
> |PseudoSeg|54.89|61.88|64.85|70.42|71|
> |PC$^2$Seg|56.9|64.63|67.62|70.9|72|
> |AugSeg|64.22|**72.17**|**76.17**|77.4|**78.82**|
> |**Ours**|**69.3**|71.02|76.09|**77.86**|78.28|
>
> >Q4. Authors adopt the U2PL splits (choose from the high-quality images first), which are unfair for some methods, such as CPS and ST++ (randomly choose from the blended set).
>
> A4. We follow the reviewer’s suggestion, and further perform experiments on ResNet-50 using the splits used in CPS and ST++ for a fair comparison.
> |Method | 1/16 (662) | 1/8 (1,323) |  1/4 (2,646) |
> |:-:|:-:|:-:|:-:|
> |CPS|71.98|73.67|74.90|
> |ST++|72.6|74.4|75.4|
> |PS-MT|72.83|75.7|76.43|
> |UniMatch|74.5|75.8|76.1|
> |AugSeg|**74.66**|75.99|77.16|
> |**Ours**|74.28|**76.15**|**77.42**|
>
> >Q5. Does the improvement come from more diverse spatial augmentations?
>
> A5. We hypothesized the truth of this question and now support it with new experimental results. We report our performance by adopting the adaptive CutMix proposed in AugSeg instead of the basic CutMix. We use PASCAL VOC 2012 with U2PL splits with ResNet-50. As shown below, we achieve mostly better results by changing the basic CutMix to a more sophisticated augmentation, adaptive CutMix. (In 1/4 partitions, we didn’t observe better performance using adaptive CutMix than basic CutMix, but tuning hyper-parameters would give better results.)
>
> |Method | 1/16 (662) | 1/8 (1,323) |  1/4 (2,646) |
> |:-:|:-:|:-:|:-:|
> |**Ours w/ CutMix**|77|79.23|**78.94**|
> |**Ours w/ Adaptive-CutMix**|**78.66**|**79.59**|78.52|
>
> >Q6. I see the authors upload the source code. Since several other works are reproduced for ADE20K, it will be more convincing and greatly appreciated if the authors can provide comprehensive experimental logs as well.
>
> A6. We followed AC's instructions and submitted the experimental logs to AC. Please check the logs from AC’s comment.

---

> > ### Comment · Reviewer_gkhQ · 2023-08-19
> > **Response to authors rebuttal**
> >
> > Thank the authors for the rebuttal.
> >
> > As for A1, please refer to my [response](https://openreview.net/forum?id=JXvszuOqY3&noteId=Xde4yUrxbV) to the rebuttal in Reviewer vns8.
> >
> > As for A2, the original method evidently lags behind AugSeg and UniMatch in most splits. Besides, these comparisons should be included in the initial submission, rather than rebuttal. During rebuttal, the newly added component Adaptive CutMix (from AugSeg) would make the whole framework much more sophisticated (more hyper-parameters) and unstable.
> >
> > As for A3, why not include UniMatch, especially since the authors have included UniMatch in A2. The results of UniMatch under the five splits are 71.9,  72.5, 76.0, 77.4, and 78.7 respectively. In comparison, the results of this submission are 69.3 (-2.6), 71.02 (-1.4), 76.09 (+0.1), 77.86 (+0.5), and 78.28 (-0.4) respectively.
> >
> > As for A4, under the same split, the improvement is very marginal, which are -0.4, +0.2, +0.2 under the three splits respectively. More importantly, I believe such fair comparisons under the same split should be included in the very initial submission, instead of reporting very mislearning results in Table 1 of the submission.
> >
> > As for A5, I did not mean trying more advanced augmentations. I hoped to see the results without using ClassMix.
> >
> > Based on the rebuttal, I would decline my score to 3.

---

> > > ### Author Response · Authors · 2023-08-19
> > >
> > > We thank the reviewer for the additional comment on our response.
> > >
> > > > [Q1] As for A1, please refer to my response to the rebuttal in Reviewer vns8.
> > > - Please refer to *our response* to your concern [here](https://openreview.net/forum?id=JXvszuOqY3&noteId=KJgF8zZ1ZE).
> > >
> > > > [Q2] As for A2, the original method evidently lags behind AugSeg and UniMatch in most splits. Besides, these comparisons should be included in the initial submission, rather than rebuttal. During rebuttal, the newly added component Adaptive CutMix (from AugSeg) would make the whole framework much more sophisticated (more hyper-parameters) and unstable.
> > > - We respectfully disagree with the reviewer’s concern that we lagged behind AugSeg (*CVPR2023*) and UniMatch (*CVPR2023*) in most splits. We followed your suggestion and reported extra results to compare them fairly. We are confident that our original method (Ours) outperformed AugSeg (e.g., in 1/8 and 1/4 in ResNet-50, and 1/4 in ResNet-101) and UniMatch (e.g., 1/8 in ResNet-50, and 1/4 partitions in ResNet-101).
> > > - Our point was not about the complexity of the method, but we just employed different augmentation, such as adaptive CutMix (dubbed Ours+). We found that ours were improved (e.g., 1/16, 1/8 in ResNet-50, and 1/4 in ResNet-101), and other results are comparable to the prior works. This experiment was *auxiliary* to this question but would partially address Q4 and Q5.
> > > - Please remember that this paper's main contribution is NOT introducing a new augmentation method. Our main contribution to this paper is *switching temporary teachers* to secure teacher diversity, a simple yet efficient method for semi-supervised learning tasks.
> > >
> > > > [Q3] As for A3, why not include UniMatch, especially since the authors have included UniMatch in A2. The results of UniMatch under the five splits are 71.9, 72.5, 76.0, 77.4, and 78.7, respectively. In comparison, the results of this submission are 69.3(-2.6), 71.02(-1.4), 76.09(+0.1), 77.86(+0.5), and 78.28(-0.4) respectively.
> > > - We did not; because the input resolution of UniMatch differs from other methods as well as ours. Most of the methods (including ours) employed *512*; UniMatch used *321* for training resolution.
> > >
> > > > [Q4] As for A4, under the same split, the improvement is very marginal, which are -0.4, +0.2, +0.2 under the three splits respectively. More importantly, I believe such fair comparisons under the same split should be included in the very initial submission, instead of reporting very mislearning results in Table 1 of the submission.
> > > - As for the marginal improvement, we gently remind you UniMatch and AugSeg are *CVPR-2023* papers. Compared with PS-MT (*CVPR-2022*), we consistently achieved 0.5-1.5% gains. Moreover, on Cityscapes, our original method achieved the best performances in all but one partition (three were the best, and one was the second-best).
> > > - We here provide a summarized table for **Cityscapes**.
> > > |Method|1/16 (186)|	1/8 (372)|	1/4 (744)|	1/2 (1,488)|
> > > |---|---|---|---|---|
> > > |PS-MT|-|76.9|77.6|79.09|
> > > |AugSeg|75.22|77.82|**79.56**|80.43|
> > > |UniMatch|76.6|77.9|79.2|79.5|
> > > |Ours|**76.81**|**78.4**|79.46 |**80.52**|
> > > - Our main contribution is introducing switching temporary teachers (i.e., teacher diversity) can improve the performance of semi-supervised learning, not simply aiming for the absolute best performance across all datasets.
> > > - UniMatch and AugSeg have significant contributions, and we recognize their importance. We believe the contributions are positively aligned with our method; as an example, we employed the adaptive CutMix utilized in AugSeg to provide an upbeat hybrid following your valuable comments. We believe Ours+ revealed room for ours' further improvements, but It is worth noting that the above table now does not include Ours+ (using adaptive CutMix).
> > > - Please note that in response to your comment to Reviewer vns8,  we corrected the performance comparison by assessing it on another split and datasets (we will revise the numbers in Table 1).
> > >
> > > > [Q5] As for A5, I did not mean trying more advanced augmentations. I hoped to see the results without using ClassMix.
> > > - We kindly remind your comment, “Does the improvement come from more diverse spatial augmentations, i.e., CutMix and ClassMix?” We understood the question was to see the applicability of further diverse augmentations, so we opted for a diverse augmentation.
> > >  - The main improvement of our method may come from teacher diversity and decoupling, as addressed in our [response](https://openreview.net/forum?id=JXvszuOqY3&noteId=MncAE7jA4h) to Reviewer vns8. In this respect, we could yield improvements by ensuring *more diversity* using adaptive CutMix with *switching temporary teaching*, as we reported in our previous response.
> > > -  We kindly remind you our method with *dual teachers employed two different augmentations*, such as CutMix and ClassMix, for sufficient diversity, so solely removing ClassMix would lose the performance; therefore used Adaptive Cutmix.

---

### Official Review · Reviewer_3Udz · 2023-07-06

**Soundness:** 3 good
**Presentation:** 3 good
**Contribution:** 3 good
**Rating:** 6
**Confidence:** 3

**Summary:**

This work proposes the dual teacher knowledge distillation algorithm. For teachers trained with EMA during KD, there exists the issue of conformation bias which has been tackled in previous works. The motivation of this work is to diversify the training strategy using the dual-teacher student algorithm, in a way that it reduces the conformation bias as much as possible. This is done by making sure the teacher and student don't converge to the same representation in the feature space, using the proposed algorithm, which is a combination of augmentations to the inputs as well as switching teachers to train a student. The results obtained are quite promising compared to some recent KD schemes.

**Strengths:**

Presentation:
The ideas are presented well. The concept is clear and explained in a way which is understandable to the reader. The reviewer appreciates the fact that figures and tables are self-explanatory, which improves the papers clarity. It reads well in a single pass.

Concept:
The idea seems novel and well-motivated by using observations from prior art[1].

Experiments:
The results are promising. It seems the main advantage of this method is that it converges faster than other related teacher-student EMA based algorithms published recently.

Transparency:
The authors have made the code available and also addressed limitations and societal impacts of their work.

[1] Pseudo-Labeling and Confirmation Bias in Deep Semi-Supervised Learning

**Weaknesses:**

One major concern is, as the authors have augmented their inputs to train the student, have they also augmented the data similarly while training other methods to compare with their proposed method.

**Questions:**

1. Is there any study to show that the performance improvements from this approach are not attributed only to the input augmentation scheme?
2. How do you tune the terms in equation 3?
3. Does the training of the student require starting from a frozen checkpoint?

**Limitations:**

Yes

---

> ### Author Rebuttal · Authors · 2023-08-09
>
> We appreciate your valuable comments and positive reviews. We address your concerns and questions below in detail.
>
> >Q1. Did the authors apply similar augmentations to other methods?
>
> A1. Yes, we utilize basic augmentations such as CutMix and ClassMix, which is a similar level of augmentation compared to recent methods. For instance, CPS, PS-MT, and U$^2$PL utilized CutMix, ReCo used ClassMix, and AEL and AugSeg applied advanced augmentations such as adaptive copy-paste CutMix and adaptive label-injecting CutMix.
>
> >Q2. Is there any study to show that the performance improvements from this approach are not attributed only to the input augmentation scheme?
>
> A2. Yes. In Table 5, we reported that not only augmentation but also our training protocols that switch the temporary teachers make a significant contribution to performance improvement. For example, our strategy of switching temporary teachers shows a performance improvement of about 2% mIoU, regardless of the number of augmentations.
>
> >Q3. How do you tune the terms in Equation 3?
>
> A3. In all experiments, we set the hyperparameter $\lambda_u$ as 1 to train the supervised and unsupervised losses in equal proportions.
>
> >Q4. Does the training of the student require starting from a frozen checkpoint?
>
> A4. Yes. In our experiment, we use ResNet-50/101 pre-trained on ImageNet as the backbones and ResNet-101 pre-trained on MS COCO for the domain adaptation task in Table 7. We employ the identical backbones utilized in the previous works.

---

> > ### Comment · Reviewer_3Udz · 2023-08-15
> >
> > Thanks for addressing my questions. I keep my original rating.

---

### Official Review · Reviewer_vns8 · 2023-07-06

**Soundness:** 2 fair
**Presentation:** 3 good
**Contribution:** 1 poor
**Rating:** 4
**Confidence:** 5

**Summary:**

This work proposes a method for semi-supervised semantic segmentation of images.
To mitigate the coupling problem of the student-teacher framework, the author proposes Dual teacher framework and  Implicit Consistency Learning.
Dual teacher framework consists of a pair of teacher-student models, where the two temporary teacher models are switched every epoch during training to teach the single student model.
In addition to this,  Implicit Consistency Learning is proposed, where the layer of the student model is stochastically dropped out (so becomes the so-called sub-model) and trains such sub-model by enforcing the prediction of the sub-model and the teacher model are same.
This approach shows the state-of-the-art performance for all dataset in every settings.

**Strengths:**

1. This paper shows the state-of-the-art performance compared to the previous and supervised methods.
2. This paper conducts thorough experiments, showing their component's effectiveness.
3. This paper is easy-to-follow and clearly motivated.

**Weaknesses:**


The major concern of the reviewer in this paper is its technical contribution.

The main idea of this paper is: to introduce two teachers and students, and each of them is updated alternately activated per epoch, e.g., for epoch t, teacher model 1 is updated by student model 1 (and t2 is updated by s2), and for epoch t+1, teacher model 1 is updated by student model 2 (and t2 is updated by s1), and so on.

However, this approach is very similar to PS-MT [1]. In PS-MT, it introduced two teacher model that is updated by one single student model.  And in PS-MT, such teacher models are updated 'alternatively' per epoch; that is, for epoch t, only one teacher network is updated while the other is frozen, and for epoch t+1, the frozen one is now updated, and the updated one is now frozen.

This training strategy is introduced to diversify the teacher model's output, which is exactly the same as the proposed method's objective.

So hereby the reviewer thinks that the proposed method is just a variant of PS-MT's method, and have a question about why the author of this paper did not compare their method with PS-MT's one.

In addition to this, the reviewer thinks that Implicit Consistency Learning is just a variant of the previous methods as the author already mentioned in their paper, and the only difference is that the source of consistency has been changed like this: from (sub-model of the student <--> full model of the student: previous method ) to (sub-model of the student <--> full model of the teacher: proposed method).

So the reviewer thinks the proposed components lack of their technical contributions at the current state of the paper.


Ref.

[1] Yuyuan Liu, Yu Tian, Yuanhong Chen, Fengbei Liu, Vasileios Belagiannis, and Gustavo Carneiro. Perturbed and strict mean teachers for semi-supervised semantic segmentation. In Proceedings of the IEEE/CVF Conference on Computer Vision and Pattern Recognition, pages 4258–4267, 2022.



**Questions:**

1. What is the difference between PS-MT and the proposed one? Please give me a detailed comparison between them.

2. Can you visualize (or numerically show) how much the proposed method diversifies teacher models' output compared to PS-MT or the typical mean-teacher model?

**Limitations:**

See weaknesses.

---

> ### Author Rebuttal · Authors · 2023-08-09
>
> We appreciate your constructive feedback. We address all the concerns raised by the reviewer and provide some new experimental results to support our contributions.
>
> >Q1. What are the main differences from PS-MT?
>
> A1. We would like to emphasize the notable differences.
>
> 1. Our approach differs from PS-MT in that only one switchable teacher model teaches a student model per epoch, and this difference allows our method to obtain a more distinct student model for each epoch.
> 2. To ensure diversity of our dual temporary teachers, we update the teacher model from a student model trained with different augmentations for each epoch.
>
> Specifically, in PS-MT, two teachers always contribute to training the student model in all epochs. They use the ensembled outputs of the two teacher models in every epoch to provide better pseudo-labels to the student model. However, this training approach is limited to obtaining a student model that varies over epochs and results in teacher models that are updated via EMA from the student model not diversifying enough.
>
> On the other hand, in our method, only one of the temporary teacher models involves teaching the student model in each epoch. This approach was intended to diversify the teacher models: our dual teachers are trained to have different perspectives using different types of augmentations, and only one of the teacher models independently teaches the student model with a distinct perspective at each epoch. We argue that ensembling teacher models ensures consistent supervision for our method but diminishes diversified guidance for the student model, which potentially hinders the student model from becoming more generalized in the end. Furthermore, we provide the numerical comparison table on this concern in the table of A4.
>
> To provide the backup for our claim, we perform the following experiments to demonstrate that ensembling the outputs of the two teacher models negatively affects obtaining diverse teacher models in our method. Here, we ensemble the outputs of the two teacher models in equal proportions, according to the ensemble method of PS-MT. Please check the results in the table below.
>
> |Method | 1/16 (186) | 1/8 (372) |  1/4 (744) | 1/2 (1,488) |
> |:-|:-:|:-:|:-:|:-:|
> |**Ours + Switching temporary teaching**|77.0|79.23|78.94|80.18|
> |**Ours + Ensemble teaching**|73.25|75.99|76.41|77.34|
>
> >Q2. Why does the author not compare their method with PS-MT?
>
> A2. We already compared our method with PS-MT on both PASCAL VOC and Cityscapes datasets and verified our method has achieved better performances compared with PS-MT. Please check Tables 1 and 3 in our submitted paper.
>
> >Q3. Is implicit consistency learning a variant of the previous methods?
>
> A3. The aspect of consistent learning between the student's sub-model and the teacher's full-model within the Mean Teacher framework has not been previously tackled. Through extensive analysis and experimental validation on our method, we believe this can make a valuable contribution in this field.
>
> >Q4. Can you show how much the proposed method diversifies teacher models' output compared to PS-MT or the typical mean-teacher model?
>
> A4. We compared our method with the typical mean-teacher model in Figure 2 (a) in the paper. Here we further compare with PS-MT in the following tables for teacher diversity. Further details will follow:
>
> 1. Comparison with typical mean-teacher:
>
> Figure 2. (a) shows the comparison results in prediction distance between typical mean-teacher (i.e., Single Teacher) and our dual teacher models. Our Dual Teacher consistently maintains a larger distance between the student and teachers than the typical mean-teacher model during training.
>
> 2. Comparison with PS-MT:
>
> We compare the output predictions of two teachers of ours and PS-MT using class-wise IoU metric, emphasizing noticeable differences. We highlight the top-5 categories with "the most significant differences ($\Delta$Diff)" in class-wise IoU between the teacher models. For a fair comparison, we compute class-wise IoU when both teachers have similar mIoU values (e.g., Ours: 66.48 / 66.01 / PS-MT: 67.09 / 67.37). The results demonstrate a larger distance and a clear distinction between teachers, indicating that our method produces more diverse teacher models than PS-MT, which ensembles teacher outputs every epoch.
>
> - Proposed method (top-5 categories with the most significant differences):
> |Category | cow | dog |  cat | bottle | sheep |
> |:-:|:-:|:-:|:-:|:-:|:-:|
> |**Teacher #1**|58.73|61.24|61.66|63.16|79.55|
> |**Teacher #2**|77.70|52.98|69.81|56.91|75.97|
> |**$\Delta$Diff**|18.97|8.26|8.15|6.20|3.58|
>
> - PS-MT (top-5 categories with the most significant differences):
> | Category | bird | plant | table | chair | motorbike |
> |:-:|:-:|:-:|:-:|:-:|:-:|
> |**Teacher #1**|79.68|49.19|47.38|20.63|73.78|
> |**Teacher #2**|84.33|45.62|49.09|22.31|75.34|
> |**$\Delta$Diff**|4.65|3.57|1.71|1.68|1.67|

---

> > ### Comment · Reviewer_vns8 · 2023-08-16
> >
> > Thank you for the authors’ response to the questions.
> >
> > However, the reviewer still has several concerns about the paper. First, the core idea of the paper seems to be just an incremental design to PS-MT, although it shows somewhat better results. Second, the reviewer is curious about the inference method. How is the performance evaluated and with which model? There might be several student and teacher models with distinct abilities, so how is the inference performed? The authors are expected to explain the procedure of inference more clearly.
> >
> > Therefore, the reviewer is considering keeping the rating as borderline reject.

---

> > > ### Author Response · Authors · 2023-08-16
> > >
> > > We appreciate your constructive feedback on this manuscript and comments on your concerns. We would like to clarify the difference between ours and PS-MT again and handle your inference-related concern.
> > >
> > > **1. Concern about the inference method.**
> > >
> > > All our performances were evaluated by **using only a single student model** without any ensemble with teacher models. We certainly acknowledged that the ensemble of teacher and student models can improve better performance, but we did not use it in this paper.
> > >
> > > **2. Our main purposes of our method are [1] establishing a *decoupling* between teacher and student, [2] promoting *diversity* between teachers, [3] simple yet *efficient perturbation*, unlike PS-MT**.
> > >
> > > > [1] Main difference between PS-MT and ours lies in the method employed for teaching a pseudo label for a student.
> > >
> > > &emsp;&emsp;PS-MT&emsp;&emsp;|&emsp;Ours\
> > > ------------------------------------\
> > > $T_1$┐Ensemble&ensp;|&emsp;$T_1$&emsp;$T_2$\
> > > $T_2$┘&emsp;&emsp;↓&emsp;&emsp;&ensp;|&emsp;↓&emsp;&emsp;↓\
> > > &emsp;&emsp;&emsp;&ensp;$S$&emsp;&emsp;&ensp;|&emsp;$S$&nbsp;&nbsp;↔&nbsp;$S$\
> > > ------------------------------------\
> > >
> > > As shown in the above figure, for establishing the *decoupling*, we switched teachers every epoch to guide the pseudo label to the student. However, PS-MT guided the pseudo label from the ensemble of teachers to the student.
> > > Specifically, our method:\
> > > [Epoch t+1 ]   $T_1$ teaches $S$ / then, $T_2$ is decoupled from $S$.\
> > > [Epoch t+2 ]   $T_2$ teaches $S$ / then, $T_1$ is decoupled from $S$.
> > >
> > > PS-MT:\
> > > [Epoch t+1 ] [$T_1$+$T_2$] teaches $S$ / then, Student cannot be decoupled from any Teachers.\
> > > [Epoch t+2 ] [$T_1$+$T_2$] teaches $S$ / then, Student cannot be decoupled from any Teachers.
> > >
> > > From this sequential switching mechanism, we can achieve the *decoupling* of the student and teachers. It results in better performance compared with PS-MT.
> > >
> > > > [2] Main difference comes from diverse augmentations for students (e.g., $SA_1$ and $SA_2$).
> > >
> > > &emsp;&emsp;&emsp;PS-MT&emsp;&emsp;&emsp;&emsp;|&emsp;&emsp;&emsp;Ours\
> > > ---------------------------------------------------------------------\
> > > (WA→$T_1$)┐Ensemble&ensp;|&emsp;(WA→$T_1$)&emsp;(WA→$T_2$)\
> > > (WA→$T_2$)┘&emsp;&emsp;↓&emsp;&emsp;&ensp;|&emsp;&emsp;&emsp;&emsp;↓&emsp;&emsp;&emsp;&emsp;&emsp;↓\
> > > &emsp;&emsp;&emsp;&emsp;(SA→$S$)&emsp;&emsp;&ensp;|&ensp;($SA_1$→$S$)→&nbsp;($SA_2$→$S$)\
> > > ----------------------------------------------------------------------\
> > >
> > > In the *diversity* of teachers, our method achieved better diversity of the teachers because of our switching teacher mechanism (supported by our different augmentations), unlike the simple ensemble of teachers. The proof of this concept has been done by the following experiments where we highlighted the top-5 categories with "the most significant differences ($\Delta$Diff)" in class-wise IoU between the teacher models.
> > >
> > > - Our method
> > > |Category | cow | dog |  cat | bottle | sheep |
> > > |:-:|:-:|:-:|:-:|:-:|:-:|
> > > |**$\Delta$Diff of $T_1$ and $T_2$**|18.97|8.26|8.15|6.20|3.58|
> > >
> > > - PS-MT
> > > | Category | bird | plant | table | chair | motorbike |
> > > |:-:|:-:|:-:|:-:|:-:|:-:|
> > > |**$\Delta$Diff of $T_1$ and $T_2$**|4.65|3.57|1.71|1.68|1.67|
> > >
> > > From this different augmentation mechanism, we can achieve the *diversity* of the dual teachers, contributing to better performance than PS-MT.
> > >
> > > > [3] Main difference in perturbation mechanism.
> > >
> > > &emsp;&emsp;&emsp;PS-MT&emsp;&emsp;&emsp;&emsp;|&emsp;&emsp;&emsp;Ours\
> > > -------------------------------------------------------------------\
> > > Feature pertubation&ensp;|&ensp; Layer pertubation\
> > > &emsp;(T-VAT-based)&emsp;&emsp;&ensp;&nbsp;|&ensp;(Stochastic depth-based)\
> > > --------------------------------------------------------------------\
> > >
> > > We employed stochastic depth-based layer perturbation to the student model for further consistency learning effect, and PS-MT used T-VAT-based feature perturbation. Specifically, our perturbation is simply made in the student model, while PS-MT's perturbation is made from the ensemble of teachers to the student. This difference allows us to achieve a simple yet *efficient perturbation* for faster training than PS-MT.

---

> > > > ### Comment · Reviewer_vns8 · 2023-08-17
> > > >
> > > > Thank you for the authors’ explanation of the questions. The reviewer acknowledges that the authors have addressed some issues, such as novelty and inference methods, and that the proposed method has some advantages over PS-MT. Therefore, the reviewer has a positive mindset and is willing to raise their initial rating from borderline reject to borderline accept. However, the reviewer have decided to wait for the other reviewers’ responses.

---

> > ### Comment · Reviewer_gkhQ · 2023-08-19
> > **Why the ensemble teaching is so poor?**
> >
> > I am Reviewer gkhQ. I am very shocked about the reported results of ensembled teaching in Q1. I cannot understand why the ensembled teaching is much poorer than the switching teaching. According to my experiments, the results of simple single EMA teaching are already much higher than the reported results.
> >
> > Take the 1/8 split as an example (btw, the authors write the wrong absolute number in the brackets), the PS-MT result is 75.7. Considering that **this submission uses very unfair labeled splits  (much higher-quality) than PS-MT** (refer to [this issue](https://github.com/Haochen-Wang409/U2PL/issues/3) for details), the PS-MT result under the same split should be nearly 78+. From my perspective, PS-MT is just a method of ensemble teaching. Therefore, I do not agree with the results that the authors provide in the rebuttal.

---

> > > ### Author Response · Authors · 2023-08-19
> > >
> > > Dear reviewer gkhQ,
> > >
> > > We sincerely thank your response, and we hope this response addresses and clarifies your concerns.
> > > - **Factor analysis of two teaching methods** - borrowed the table from our previous response
> > >   |Method| 1/16 (662)|1/8 (1,323)|1/4 (2,646)|	1/2 (5,291)|
> > >   |---|---|---|---|---|
> > >   |Ours + Switching temporary teaching|	77.0| 79.23| 78.94| 80.18|
> > >   |Ours + Ensemble teaching| 73.25| 75.99| 76.41| 77.34|
> > >
> > > > On the study comparing switching teachers vs. ensembled teaching (we show the previous table once more for better readability)
> > >
> > > - We first correct the numbers in the partitions for the high-quality set in PASCAL VOC 2012 with 662, 1,323, 2,646, and 5,291.
> > >
> > > - We would like to clarify that
> > >
> > >   - These experiments were performed without using T-VAT and other crucial components of PS-MT *to solely compare the impact of the two methods*. Therefore, ensemble teaching may not reach the performance you expected.
> > >   - We argue that the contribution of PS-MT is **not** just a method of ensemble teaching. Specifically, Table 5 in the PS-MT paper reported significant improvements with *components such as conf-CE (new loss), T-VAT, and AT (auxiliary teacher)*.
> > >   - We gently remind you that *the above results were produced on our framework between switching temporary teaching and ensemble teaching for a fair comparison, not on PS-MT that leverages the aforementioned key components.* Namely, they are a factor comparison of two teaching methods.
> > >
> > > - Our performance improvements presumably stem from **teacher diversity**. We left the analysis to testify for Reviewer WnDT. Please see this link (https://openreview.net/forum?id=JXvszuOqY3&noteId=M18pP5G1ug)
> > >
> > > > We respectfully disagree with Reviewer gkhQ regarding the assertion of an unfair comparison with PS-MT:
> > > - We already provided a fair comparison between PS-MT and ours according to your suggested PASCAL VOC split in our previous response (A4 at https://openreview.net/forum?id=JXvszuOqY3&noteId=fCRoTUBZOQ). We now offer the following summarized comparison table for clarification.
> > >   - **Reviewer gkhQ 's suggestion of PASCAL VOC split** (we summarize performances only between ours and PS-MT here.)
> > > |Method| 1/16 (662) |1/8 (1,323) |1/4 (2,646)|
> > > |---|---|---|---|
> > > |PS-MT |72.83 |75.7| 76.43|
> > > |Ours |**74.28**| **76.15** |**77.42**|
> > >
> > > - Furthermore, in the first submission, we also compared with PS-MT on CityScape in Table 3, and we observed that ours evidently outperformed PS-MT for all partitions.
> > >   - **Cityscapes** (here we show a summarized performance between ours and PS-MT.)
> > > |Method| 1/16 (186)| 1/8 (372)| 1/4 (744)| 1/2 (1,488)|
> > > |---|---|---|---|---|
> > > |PS-MT| - |76.89| 77.60 |79.09|
> > > |Ours| 76.81| **78.4**| **79.46**| **80.52**|
> > >
> > > - From two fair comparison tables from PASCAL VOC and Cityscape evaluation protocols, we can observe that our method achieved better results than PS-MT.
> > >
> > > - Even though PS-MT reaches 78+ accuracy as claimed by Reviewer gkhQ, ours surpassed 79+ (i.e., 79.23) at 1/8 split. Therefore, these results (including the above two results) and our results support the effectiveness of our method.

---

### Decision · Program_Chairs · 2023-09-21

**Decision:**

Accept (poster)

**Comment:**

This paper received mixed reviews: two weak accepts, one borderline reject, and one reject. The reviewers appreciated strong performance, extensive experiments, and clarity of the draft. They at the same time raised concerns with limited technical contribution (vns8, gkhQ), lack of comparisons with latest work (gkhQ), unfair comparisons with some previous work (gkhQ), missing ablation studies (gkhQ, WnDT), and lack of reasoning for experimental results (WnDT).

The authors provided thorough rebuttals that address these comments; most of the concerns have been well assuaged, but a few remain unresolved, e.g., missing details of inference given two student models (vns8) and potential adverse effects by the specialization of teachers (vns8). Also, a reviewer expressed disappointment, questioning why the comparisons with latest methods (presented in CVPR 2023) were not included in the initial submission (gkhQ). Hence, even after a large body of discussions, the scores still diverge.

The AC found that the concerns on missing details of inference and potential adverse effects of the proposed decoupling strategy can be addressed by simple additional experiments or already have been addressed by existing results. Further, these concerns were raised after the author-reviewer discussion period and thus the authors did not have a chance to address them unfortunately. More importantly, comparisons with CVPR 2023 papers (i.e., UniMatch and AugSeg) are not mandatory since they were officially presented after the submission deadline of NeurIPS 2023; marginal improvement over or scores inferior to those of the CVPR 2023 papers should not be a factor for rejection.

Putting these together, the AC considers that these concerns are minor or inappropriate, and thus the strengths and the rebuttal outweigh the negative aspects of the paper. Hence, the AC recommends accepting the paper even though the average score is not sufficiently high; this decision has been extensively discussed between the AC and SAC. The authors are strongly encouraged to reflect the valuable comments and suggestions by the reviewers, and to include additional results given in the rebuttals, in the revision.